# Retrieval of cloud thermodynamic phase partitioning from multi-angle polarimetric imaging of Arctic mixed-phase clouds

Anna Weber[1], Veronika Pörtge[1], Claudia Emde[1,a], and Bernhard Mayer[1]

[1]Meteorologisches Institut, Ludwig-Maximilians-Universität München, Munich, Germany
[a]now at: Rayference, Bruxelles, Belgium

**Correspondence:** Anna Weber (Weber.Ann@physik.uni-muenchen.de)

**Abstract.** Mixed-phase clouds are frequently observed in the Arctic and still not well represented in climate and general circulation models. The spatial distribution of cloud thermodynamic phase and its partitioning are important quantities since they affect the radiative effect of clouds as well as cloud life time. In this work, a new quantitative retrieval method of cloud thermodynamic phase partitioning based on multi-angle polarimetry is presented. The polarization signal is sensitive to cloud thermodynamic phase since liquid water and ice have different shapes and different optical properties. The basic idea of the retrieval is to fit simulations obtained from a forward operator to measurements in the cloudbow range between $135°$ and $165°$ scattering angle and the slope range between $60°$ and $110°$ to determine a quantitative ice fraction. Either plane-parallel clouds or three-dimensional (half-spherical) clouds are assumed in the simulations. The retrieval uses data measured by the polarization-resolving cameras of the specMACS instrument and provides two-dimensional fields of cloud thermodynamic phase partitioning with a high spatial resolution of about $100\,\mathrm{m}$. The retrieval was validated using synthetic data. 3D radiative transfer simulations were performed for different idealized cloud cases as well as for a realistic field of low-level Arctic mixed-phase clouds. In addition, it was applied to measurements taken during the HALO–$(\mathcal{AC})^3$ campaign. As the retrieval is based on polarization, it is sensitive to cloud top. The retrieved ice fraction is here defined as the ratio of the ice optical thickness to the liquid plus ice optical thickness and corresponds to the mean ice fraction of the uppermost cloud layer from cloud top to an optical thickness of about 1 to 2, depending on the solar zenith angle.

## 1 Introduction

Mixed-phase clouds are characterized by the coexistence of supercooled liquid water and ice (Korolev et al., 2017). They are observed at temperatures between the threshold for homogenous freezing at about -40 °C and the melting temperature of ice at 0 °C. In the Arctic, mixed-phase clouds are particularly important and observed in 40 % to 70 % of the time with the highest frequencies in spring and fall (Shupe et al., 2006; Shupe, 2011). Despite the inherent instability of the mixed-phased state, Arctic mixed-phase clouds persist on average 12 hours and occasionally up to several days (Shupe et al., 2006; Morrison et al., 2012). A thin liquid layer is usually located at the cloud top and ice crystals form from this layer through heterogeneous nucleation and sediment downwards. The ice crystals in low-level Arctic mixed-phase clouds can grow rapidly at the expense of the supercooled liquid via the Wegener-Bergeron-Findeisen process (Morrison et al., 2012). The geometrical thickness of

these clouds is typically between a few hundred meters to about 1.5 km (Shupe et al., 2006), and the geometrical thickness of the supercooled liquid layer is on the order of 100 m (Schirmacher et al., 2024). Typical liquid water paths and ice water paths are on the order of $100\,\mathrm{g m^{-2}}$ and $10\,\mathrm{g m^{-2}}$, respectively (Shupe et al., 2006). The partitioning and spatial distribution of cloud thermodynamic phase are important quantities since cloud thermodynamic phase affects cloud cover and cloud life time (Pithan et al., 2014). Moreover, liquid water and ice have different optical and radiative properties and cloud phase partitioning thus

determines the radiative effect of clouds (Choi et al., 2014; Matus and L'Ecuyer, 2017), which also impacts surface processes and sea ice extent (Morrison et al., 2012).

Comparisons between observations and model simulations showed that mixed-phase clouds and their microphysics are not well represented in climate and general circulation models (Morrison et al., 2012; Pithan et al., 2014; Komurcu et al., 2014; Cesana et al., 2015, 2022). In addition, there are uncertainties in the projections of future climate due to the representation

of mixed-phase clouds, especially in the Arctic (Tan and Storelvmo, 2019). Future climate projections with an improved representation of mixed-phase clouds recently showed increased global warming and an underestimation of climate sensitivity (Tan et al., 2016; Hofer et al., 2024). Thus, further observations of cloud thermodynamic phase are crucial to improve the process understanding of mixed-phase clouds and provide constraints to models.

Existing retrievals of cloud thermodynamic phase from passive remote sensing use spectral absorption differences between

liquid water and ice in the near-infrared. Based on this, Ehrlich et al. (2008) and Jäkel et al. (2013) defined so-called ice indices to distinguish between the liquid, mixed, and ice phases. Thompson et al. (2016) extended the spectral approach by introducing an improved spectrum fitting method and Ruiz-Donoso et al. (2020) studied the horizontal distribution of the ice index in Arctic mixed-phase clouds and combined it with radar measurements. In addition, Wang et al. (2020) developed a machine learning approach with a random forest for phase classification using Meteosat Second Generation data. Cloud thermodynamic phase

was also derived from polarization measurements of POLDER (Goloub et al., 2000) and combined with the spectral retrieval of cloud phase from MODIS to classify clouds into liquid, mixed, and ice clouds (Riedi et al., 2010). Active remote sensing using lidar and radar measurements, on the other hand, allows to study the vertical structure of cloud thermodynamic phase by using a synergistic retrieval as in e.g. Mioche et al. (2015). Finally, Mayer et al. (2024) followed a probabilistic approach using passive satellite measurements to classify clouds into mixed, ice, supercooled liquid, or warm liquid. The mentioned retrieval

methods, however, do not provide quantitative phase information about the partitioning between liquid water and ice, and the spectral retrievals are strongly affected by 3D radiative effects, especially at large solar zenith angles, which are typical in polar regions.

In-situ observations of mixed-phase clouds and cloud thermodynamic phase were analyzed by e.g. McFarquhar et al. (2007); Jourdan et al. (2010); Mioche et al. (2017), and Moser et al. (2023). In contrast to the retrievals based on remote sensing, in

situ measurements provide quantitative information about cloud phase partitioning, but provide information only along the one-dimensional flight track. In addition, accurate phase discrimination in mixed-phase clouds is also challenging for in situ measurements (Korolev et al., 2017). In order to obtain accurate quantitative results, a combination of different in situ probes is necessary. Cloud phase partitioning can be determined either through measurement of the liquid and total water content or

through the asymmetry parameter, but the measurements usually have large uncertainties. For example, Moser et al. (2023) found that the uncertainty for measurements of the cloud water content in mixed-phase clouds ranges from 20 to 50 %.

In this work, a new quantitative retrieval method of cloud thermodynamic phase partitioning using multi-angle polarimetric imaging is presented. Liquid water and ice have different shapes and different optical properties, which results in different polarization signals. Using the polarization signal is therefore a possibility to estimate the cloud thermodynamic phase, which is representative for the cloud top. The basic idea of the retrieval is to fit simulated profiles of the second Stokes vector component $Q$ (describing linear polarization) along the scattering plane to measured profiles of $Q$ to determine the ice fraction, which is here defined as the ratio of the ice optical thickness to the total optical thickness. For the optimization, the neural network forward operator for polarized 3D radiative transfer by Weber et al. (2025) is used, assuming either plane-parallel clouds or using the InDEpendent column local halF-sphere ApproXimation (IDEFAX), where the 3D cloud geometry is accounted for. The retrieval is validated using synthetic measurements computed with the Monte Carlo radiative transfer solver MYSTIC (Mayer, 2009; Emde et al., 2010) operated in libRadtran (Mayer and Kylling, 2005; Emde et al., 2016) for idealized cloud cases and for realistic low-level Arctic mixed-phase clouds, which were simulated with the Weather Research and Forecasting (WRF) model (Skamarock et al., 2019). Moreover, the retrieval is applied to measurements of the 2D RGB polarization-resolving cameras of the specMACS instrument (Ewald et al., 2016; Weber et al., 2024) taken during the HALO–$(\mathcal{AC})^3$ campaign (Wendisch et al., 2024; Ehrlich et al., 2025).

The paper is structured as follows. The specMACS instrument is described in Sect. 2 and the retrieval of cloud thermodynamic phase partitioning is explained in Sect. 3. Next, the vertical signal location of the retrieved ice fraction is characterized in Sect. 4. The retrieval is validated with synthetic data in Sect. 5 and an application to measurement data is shown in Sect. 6 followed by a summary in Sect. 7.

## 2 Instrument description

The specMACS instrument is a hyperspectral and polarization-sensitive imaging system that consists of two spectrometers for the visible and near-infrared wavelength range (Ewald et al., 2016) and two 2D RGB polarization-resolving cameras (Weber et al., 2024). It is operated in a downward-looking perspective on board the German research aircraft HALO. The polarization-resolving cameras have a large maximum combined field of view of $91° \times 117°$ in along-track and across-track direction and measure with 8 Hz acquisition rate. Moreover, the cameras provide measurements of absolute calibrated Stokes vectors $(I, Q, U)$ in red, green, and blue color channels and have a high spatial resolution of about 10 m for a typical flight altitude of 10 km. A detailed description of the instrument characteristics can be found in Weber et al. (2024). While flying above a scene, identified targets on a cloud are observed from different viewing angles, which allows to apply multi-angle polarimetric methods. Aggregated signals with the different viewing angles for a given cloud target have a spatial resolution of approximately 100 m depending on the distance between the airplane and the cloud. Existing retrievals so far include the stereographic retrieval of 3D cloud geometry by Kölling (2020), which does not use polarization information, and the retrieval of cloud droplet size distribution from cloudbow observations by Pörtge et al. (2023).

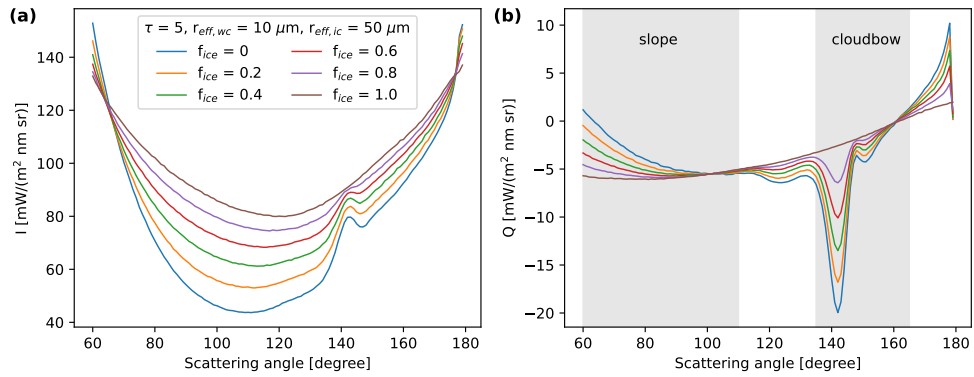

**Figure 1.** Simulation of $I$ (a) and $Q$ (b) along the scattering plane for a homogeneously mixed plane-parallel cloud with a total optical thickness 5 and varying ice fractions. The simulations were performed for 550 nm wavelength, a solar zenith angle of $70°$, and using the ice optical properties by Yang et al. (2013) as implemented by Emde et al. (2016) for moderately rough 8-column aggregates. Effective radii for liquid and ice clouds were 10 μm and 50 μm, respectively.

## 3 Retrieval description

As discussed by Goloub et al. (2000), the polarization signal is sensitive to cloud thermodynamic phase since liquid water and ice have different optical properties. While liquid cloud droplets are spherical, ice crystals have irregular shapes with varying complexities and thus different (polarized) scattering phase functions. Figure 1 shows radiative transfer simulations of the Stokes vector components $I$ (a) and $Q$ (b) along the scattering plane for a homogeneously mixed plane-parallel cloud with a total optical thickness of 5 and varying ice fractions. The ice fraction $f_{\text{ice}}$ is here defined as the fraction of the ice optical thickness to the total optical thickness:

$$f_{\text{ice}} = \frac{\tau_{\text{ice}}}{\tau_{\text{liquid}} + \tau_{\text{ice}}}. \tag{1}$$

The simulations were performed using the library for radiative transfer libRadtran (Mayer and Kylling, 2005; Emde et al., 2016) and the Monte Carlo solver MYSTIC (Mayer, 2009; Emde et al., 2010; Buras and Mayer, 2011). The shown simulations are for a wavelength of 550 nm, which is close to the center wavelength of the green color channel of the polarization-resolving cameras of specMACS and was used here as a representative wavelength. Simulations for the broader color channels of spec-MACS look very similar. Focusing on the polarization signal $Q$ in Fig. 1b, there are three angular ranges that are sensitive to cloud thermodynamic phase partitioning. In the cloudbow range from $135°$ to $165°$ scattering angle, the amplitude of the minimum at around $140°$ scattering angle decreases with increasing ice fraction until no minimum is visible anymore. This minimum is the so-called cloudbow, which is formed by scattering on (spherical) liquid cloud droplets. Observation of the cloudbow generally indicates the presence of liquid water and absence of the cloudbow a pure ice cloud. The cloudbow is also strongly sensitive to the cloud droplet size distribution and the effective radius and variance of the cloud droplet size distribution can be derived from the measurements using the cloudbow retrieval by Pörtge et al. (2023). Similarly, the glory

close to 180° backscatter direction is caused by scattering on liquid cloud droplets and is not visible for pure ice clouds. The glory angular range is, however, outside the field of view of the specMACS instrument for large solar zenith angles as in the Arctic and covers only a small angular range. Therefore, the glory signal is not used for the retrieval. Finally, the slope of the polarization curve changes depending on the partitioning between liquid water and ice in the angular range between 60° and 110°, which we therefore call slope range. Increasing ice fractions lead to a less negative slope.

The basic idea of the retrieval is to fit simulated profiles of $Q$ as in Fig. 1b to measured profiles, similarly to the cloudbow retrieval of cloud droplet size distribution. Depending on the viewing and solar geometry, this fit is done for the cloudbow range or the slope range. An overview of the different retrieval steps is given in Fig. 2 and the individual retrieval steps are explained in more detail in the next sections. First, the stereographic retrieval of 3D cloud geometry by Kölling et al. (2019) is applied to the measurements. The obtained information is then used to identify cloud targets and geolocate them to compute so-called level 1C (L1C) data, which is aggregated signals of observations at different scattering angles for the individual targets (Pörtge et al., 2023). In case the cloudbow range is covered, the cloudbow retrieval by Pörtge et al. (2023) is applied to determine the effective radius and variance of liquid cloud droplets. After these preparatory steps, the actual phase partitioning retrieval starts. At the beginning, a simple cloudbow detection is performed. If the cloudbow is geometrically possible but not visible, the cloud is assumed to consist of pure ice and the ice fraction is assumed to be equal to 1. If a cloudbow is detected, the cloud is either a liquid or a mixed-phase cloud and the phase partitioning is retrieved in step 4. Here, cases where the polarization signal of $Q$ is saturated or not saturated are distinguished (see Sect. 3.4). Saturated refers here to a cloud with an optical thickness larger than about 3 to 5, such that the polarization signal is independent of the cloud optical thickness. In the former case, the ice fraction is directly derived from a fit of $Q$. In the latter case, the polarization signal is also dependent on the optical thickness and a combination of $I$ and $Q$ is used to derive both ice fraction and optical thickness. For the optimization, the neural network forward operator for polarized 3D radiative transfer by Weber et al. (2025) is used, assuming either one-dimensional clouds, as it is done in most retrievals, or the IDEFAX accounting for 3D cloud geometry. In the IDEFAX, the 3D cloud geometry is approximated through an independent column approximation, where each column is represented by an independent field of half-spherical clouds. The field of half-spherical clouds is defined by the local surface orientation zenith and azimuth angles and the cloud fraction. The application of the IDEFAX is possible since information about 3D cloud geometry is available from the stereographic retrieval. More details will be given below. Finally, a geometric shadow mask is applied to the results since the cloudbow retrieval and the retrieval of cloud phase partitioning are not accurate in shadow regions. All steps 1 to 5 are described in more detail in the following Sect. 3.1 to 3.5. The retrieval can, in general, be applied to the red, green, and blue color channels of the polarization-resolving cameras of specMACS. However, in the following sections of this paper, only results for the green color channel are shown, which should be preferred due to the smaller influence of Rayleigh scattering at this wavelength range and the higher spatial resolution of the measurements for this channel (Weber et al., 2024). The results for the other channels look similiar.

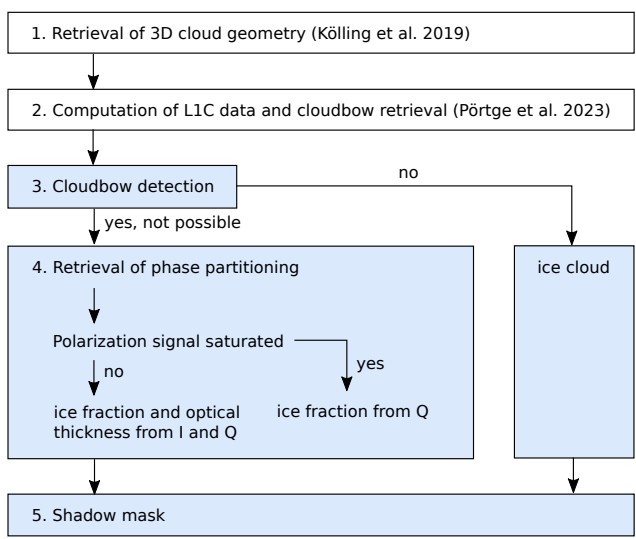

**Figure 2.** Overview of the different retrieval steps.

### 3.1 Retrieval of 3D cloud geometry

The first step of the retrieval is the stereographic reconstruction of 3D cloud geometry applying the method by Kölling et al.

(2019). The stereographic retrieval detects features with high contrast in measurements of the unpolarized intensity. While flying above a scene, the same features are viewed from different perspectives, which allows for triangulation and the derivation of the location of the features in 3D space. For this, features are identified and matched in successive images such that, in the end, a point cloud describing the cloud surface in 3D space is obtained. A mesh of triangular surfaces describing the cloud surface can then be constructed from the point cloud through Poisson surface reconstruction as described in Kölling (2020).

This triangular surface mesh is used for the geometric shadow mask in Sect. 3.5 and to compute cloud surface orientation angles for every identified cloud target for the parameterization of 3D cloud geometry in the IDEFAX forward operator. For each observed target on a cloud, the closest point on the surface mesh and the corresponding surface normal of the mesh are computed. The orientation zenith and azimuth angles are then derived from the surface normal, the local zenith, and the vector pointing towards the sun. A detailed discussion of the uncertainties of this retrieval can be found in Volkmer et al. (2024).

### 3.2 Computation of L1C data and cloudbow retrieval

In the second step, L1C data is computed from the measurements with the method by Pörtge et al. (2023), which was extended from the cloudbow range to cover the complete available scattering angle range. Here, L1C data means aggregated signals of different target points on the cloud for all observed viewing angles. For this, clouds are detected with a cloud mask and cloud targets are identified. Geolocalization of the targets is then possible using the cloud top heights from the stereographic retrieval

of cloud geometry and the known viewing geometry. Finally, signals of the individual cloud targets for different viewing angles

are aggregated from successive images. We have decided to process the multi-angle signals to an angular resolution of $0.3°$ inside the cloudbow range in order to resolve the large gradients in the cloudbow range and $1°$ otherwise.

In addition, the cloudbow retrieval of effective radius and variance of the cloud droplet size distribution (Pörtge et al., 2023) is performed if the cloudbow range is covered. The cloudbow fit uses the scattering angle range from $135°$ to $165°$ and is applied if at least the minimum cloudbow range from $136.9°$ to $160°$ (Shang et al., 2015) is observed. For the computation of L1C data, the temporal resolution of the measurements was reduced to 4 Hz, which is sufficient for the given angular resolution and reduces computation time.

### 3.3 Cloudbow detection

The first part of the actual thermodynamic phase retrieval is the cloudbow detection, whose aim is to provide a pre-selection of the data. The basic idea of the cloudbow detection is to check whether a minimum is present in the polarization signal in the cloudbow range (see Fig. 1b). The cloudbow is formed by scattering on spherical liquid water droplets and is thus an indication of cloud thermodynamic phase. If the observation of the cloudbow is geometrically possible and the cloudbow is visible, one can conclude that the cloud is either a liquid or a mixed-phase cloud. Otherwise, it is assumed to be a pure ice cloud and the ice fraction $f_{\mathrm{ice}} = 1$. In case the geometry does not allow for observing the cloudbow, no prior information about the thermodynamic phase of the cloud is available and the retrieval of cloud phase partitioning is continued with the next step. Here, the cloudbow was considered to be geometrically possible if at least $135°$ to $140°$ scattering angle were observed, such that at least a large part of the minimum is covered. To decide whether a cloudbow is detected, the minimum of $Q$ between $135°$ and $165°$ or the maximum observed scattering angle is computed. Then, the maximum between $135°$ and the scattering angle of the minimum is determined. If the difference between the maximum and minimum is larger than the average standard deviation of the measured signal, the observation is classified as cloudbow.

In order to quantify the accuracy of the cloudbow detection method, 2000 measured signals were labeled manually and the confusion matrix was computed from the manual labels and the classification from the cloudbow detection described above. To reduce the error of the derived ice fractions, the number of signals with cloudbow that are falsely classified as no cloudbow should be minimized, since this would lead to an overestimation of the ice fraction. This is the reason for the comparable generous choice of one standard deviation difference for detecting a cloudbow. If too many signals without cloudbow are falsely classified as cloudbow, the computation time of the retrieval increases, but the accuracy of the retrieved ice fraction is not affected, since the ice fraction is in this case determined by the phase partitioning retrieval in the next step. From the 1688 signals with cloudbow in the dataset, 4 % were falsely classified as no cloudbow, whereas 23.4 % of the signals without cloudbow were detected as cloudbow. During the manual labeling, it was sometimes difficult to decide whether a cloudbow was still visible or not. To reduce personal biases, the manual labeling was done by different people. Nevertheless, the manual labelling is affected by measurement noise and human interpretation. The determined accuracy of the cloudbow detection should, therefore, be interpreted as a rough estimate to prove the general applicability of the introduced cloudbow detection method. A detailed uncertainty assessment of the entire phase retrieval is carried out later in Sect. 5.

### 3.4 Retrieval of cloud phase partitioning

The core of the phase retrieval is the retrieval of cloud phase partitioning in step 4. The basic idea is to fit radiative transfer simulations to measured multi-angle signals of $I$ and $Q$ as in Fig. 1. Depending on the viewing geometry, this fit is done either for the cloudbow range for scattering angles between $135°$ and $165°$ or for the slope range between $60°$ and $110°$ scattering angle. The minimum angular range that must be observed for the cloudbow range is from $136.9°$ to $160°$ such that the cloudbow retrieval can be applied. For the slope range, scattering angles between at least $80°$ and $110°$ scattering angle must be measured since the derived ice fractions become very uncertain for smaller angular ranges as the sensitivity decreases (see Fig. 1) and smaller scattering angles are not always observed due to the viewing geometry of the instrument.

A simple lookup table containing the radiative transfer simulations is not possible due to the large number of parameters the measurements are sensitive to. These include the viewing and solar geometry (solar zenith angle, viewing zenith and azimuth angle, sensor height) which are known from the aircraft data, as well as cloud top height and cloud surface orientation zenith and azimuth angles obtained from the stereographic retrieval. Additional parameters are the effective radius and variance of the cloud droplet size distribution from the cloudbow retrieval, and the cloud fraction needed for the IDEFAX, which is computed using the cloud mask by Pörtge et al. (2023). Unknown parameters are the total cloud optical thickness and the ice fraction. Hence, the neural network forward operator for polarized 3D radiative transfer by Weber et al. (2025) was used for the fit. The neural networks predict $I$ and $Q$ converted to reflectivity for the different color channels of the polarization-resolving cameras of specMACS under the assumption of plane-parallel clouds or using the IDEFAX with half-spherical clouds. It was shown by Weber et al. (2025) that the complex 3D cloud geometry of low-level Arctic mixed-phase clouds can effectively be approximated through a half-spherical cloud that is defined by a surface orientation zenith and azimuth angle and embedded in a cloud field with a specified cloud fraction. This approximation is based on the independent column approximation and was named InDEpendent column local halF-sphere ApproXimation (IDEFAX). It is implemented in the neural network forward operator together with the classical plane-parallel independent column approximation. Depending on the observed cloud type, generally either the application of the plane-parallel assumption or the IDEFAX will be more appropriate. Moreover, the neural networks assume an ice effective radius of $50\,\mu m$, the habit of the 8-column aggregates with moderate roughness from Yang et al. (2013), and the ocean reflection function by Cox and Munk (1954a, b) combined with the Fresnel reflection matrix to include polarization. The default wind speed if no other information is available is $10\,m/s$. The wind speed determines the shape of the sunglint and, therefore, might influence the measurements. However, due to the large solar zenith angles in the Arctic, the sunglint is outside the field of view of the instrument. Mixed-phase clouds are assumed to be homogeneously mixed in the forward operator. Thus, the retrieved ice fraction has to be interpreted as an effective ice fraction of the upper most part of the cloud under the assumption of a homogeneously mixed cloud. The influence of this assumption and the vertical attribution of the retrieved ice fraction will be further discussed in Sect. 4 and 5.3.

$I$ and $Q$, of course, also depend on the optical thickness. Figure 3 shows simulation results for a homogeneously mixed cloud with a constant ice fraction of 0.5 for varying optical thickness values. In contrast to $I$, which still grows for an optical thickness of 10, $Q$ saturates quickly with increasing optical thickness. If the polarization signal is saturated, it is independent

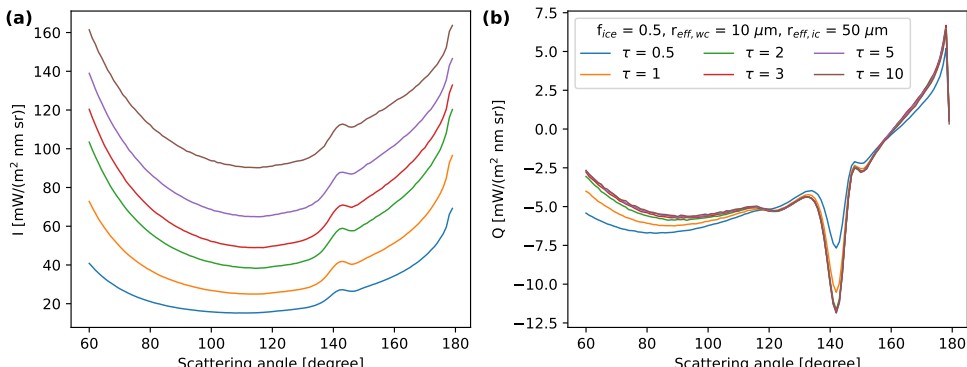

**Figure 3.** Simulation of $I$ (a) and $Q$ (b) along the scattering plane for a homogeneously mixed plane-parallel cloud with an ice fraction of 0.5 and varying total optical thickness. The simulations were performed for 550 nm wavelength, a solar zenith angle of $70°$, and using the ice optical properties by Yang et al. (2013) as implemented by Emde et al. (2016) for the habit column_8elements with moderate roughness. Effective radii for liquid and ice clouds were $10\,\mu$m and $50\,\mu$m, respectively.

of the optical thickness and the ice fraction can directly be derived from $Q$. Otherwise, $I$ and $Q$ have to be combined to simultaneously derive the optical thickness and ice fraction. The uncertainties of the derived ice fraction in this case are larger

as $I$ is more strongly affected by 3D radiative effects than $Q$ and both have different penetration depths, as will be discussed in Sect. 4 and 5. To decide whether the signal is saturated or not, measurements of the total intensity $I$ are compared to simulations with the extreme assumption of $f_{\text{ice}} = 1$, since ice clouds are brighter than liquid clouds within this range of scattering angles, in our simulated cases (see Fig. 1a). The measured polarization signal is considered saturated if the measurement of $I$ of the closest to nadir direction is larger than the calculated $I$ for an optical thickness threshold value of 5. This very conservative

threshold value ensures that unsaturated signals are not misclassified as saturated signals, which would introduce errors in the derived ice fraction.

For saturated polarization signals, the ice fraction is derived by minimizing the root mean squared error (RMSE) between modeled multi-angle signals of $Q$ and measurements of $Q$ for the respective scattering angle range:

$$\text{RMSE}_{\text{Q}}[f_{\text{ice}}] = \sqrt{\frac{1}{n}\sum_{i=1}^{n}(R_{\text{Q,meas}}(\theta_i) - R_{\text{Q,model}}[f_{\text{ice}}](\theta_i))^2}, \tag{2}$$

where $\theta$ is the scattering angle and $I$ and $Q$ are converted to reflectivity as in Weber et al. (2025), since the neural network forward operator provides these reflectivities.

If the polarization signal is not saturated, total optical thickness and ice fraction are optimized simultaneously using $I$ and $Q$ by minimizing the combined RMSE of $I$ and $Q$:

$$\text{RMSE}[f_{\text{ice}}, \tau] = (\text{RMSE}_{\text{I}}[f_{\text{ice}}, \tau] + \epsilon) \cdot (\text{RMSE}_{\text{Q}}[f_{\text{ice}}, \tau] + \epsilon), \tag{3}$$

where $\epsilon = 1 \times 10^{-4}$ is a small number. $I$ and $Q$ are not independent. Different error metrics for the optimization were tested, including commonly used weighted sums of the RMSE of $I$ and $Q$. The product form for the combined RMSE was finally chosen since it showed the best results. Nevertheless, other improved optimization metrics could be tested and incorporated in the future.

Different non-linear optimization methods, including a global search and gradient-based minimization methods, were tested. Neural networks are, by definition, differentiable and gradients can be computed through backpropagation. Thus, gradient-based optimization methods could, in principle, be applied without having to compute numerical gradients. Nevertheless, in the case of the phase partitioning retrieval, the global search was more computationally efficient. The global search was implemented as a coarse search followed by a fine search with final step sizes of 0.01 and 0.02 for the ice fraction in the saturated and unsaturated cases, respectively, and 0.2 for the optical thickness. The reason for the coarser resolution in the unsaturated case is the higher computation cost due to the combined retrieval of optical thickness and ice fraction and increased uncertainties of the results from the combined retrieval due to a larger influence of, e.g., 3D radiative effects.

In summary, the phase partitioning retrieval can be performed either for the cloudbow angular range or the slope angular range, depending on the viewing geometry. For saturated polarization signals, the ice fraction is directly derived from $Q$, whereas a combination of $I$ and $Q$ is used to retrieved the ice fraction and optical thickness for unsaturated polarization signals. Depending on the morphology of the observed clouds, the forward operator assuming plane-parallel clouds or using the IDEFAX, approximating 3D cloud geometry through a field of half-spherical clouds, can be applied.

### 3.5 Shadow mask

Finally, a geometric shadow mask is applied to the retrieval results since both the cloudbow retrieval as well as the phase partitioning retrieval are influenced by shadows, which affect the measured signals. The geometric shadow mask is computed with the surface meshes from the stereographic retrieval, similarly to Kölling (2020). A cloud target is inside a shadow if the vector from the target point on the cloud to the sun intersects the cloud surface mesh. Intersections within $100\,\mathrm{m}$ distance to the target point are not counted as intersections for a shadow. Such a geometrical shadow mask is more accurate compared to shadow masks based on a brightness threshold since the brightness of a signal is influenced by the solar geometry, different 3D radiative effects due to cloud geometry, and variations of cloud microphysics, whereas the stereographic retrieval provides accurate information about 3D cloud geometry (Volkmer et al., 2024). The shadow mask could also be applied before the phase partitioning retrieval to further reduce computation time.

### 4 Vertical attribution of the retrieved ice fraction

Before the retrieval is validated with synthetic data, the representative altitude of the signal respectively the retrieved ice fraction will be analyzed. Passive retrievals, such as the presented polarimetric retrieval of cloud phase partitioning, do not provide information about the vertical profile of the retrieved quantity. Thus, assumptions have to be made for the vertical profile and the retrieved quantities have to be referred to as effective quantities under the respective assumption. For the phase

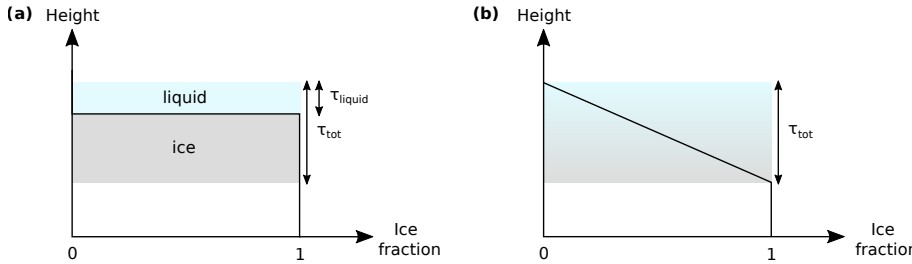

**Figure 4.** Definition of the two-layer cloud (a) and the linear profile cloud (b). The black line indicates the ice fraction at a given height level.

partitioning in low-level Arctic mixed-phase clouds, we assume two possible extreme cases for the assumption of the vertical profile. One is a completely homogeneously mixed cloud, as discussed before, the other one is a cloud with a pure liquid layer on top of a pure ice layer. The forward operator used for the retrieval assumes homogeneously mixed clouds. Moreover, the ice fraction is obtained from the polarization signal, which is dominated by the first scattering order and thus representative for the cloud top. Hence, the retrieved ice fraction is defined here as the effective ice fraction of the uppermost cloud layer from cloud top up to a certain optical thickness threshold under the assumption of a homogeneously mixed cloud.

To find this effective optical thickness threshold, respectively, the cloud depth to which the retrieval is sensitive to, simulations of a plane-parallel two-layer cloud and a plane-parallel cloud with a linear ice fraction profile in the vertical direction (see Fig. 4) were simulated and the retrieval was applied to the simulations. The two-layer cloud consisted of a plane-parallel liquid cloud with varying optical thickness ($\tau_{\mathrm{liquid}}$) above an ice cloud as visualized in Fig. 4a. The total optical thickness ($\tau_{\mathrm{tot}}$) of the cloud was 5, such that the polarization signal can be assumed to be saturated. The effective ice fraction of the cloud was retrieved with the forward operator for plane-parallel clouds for the cloudbow and the slope range for every simulated liquid cloud optical thickness. Figure 5 shows the retrieved ice fractions for a solar zenith angle of 75° as in the synthetic data used in Sect. 5.3 as a function of the liquid cloud optical thickness. From the obtained ice fractions, the optical thickness of the cloud layer to which the retrieval is sensitive to could be determined. For a liquid cloud with an optical thickness larger than about 2, the ice cloud below is not detected anymore and the retrieved ice fraction is close to 0. The vertical optical thickness threshold was then defined through the 90th percentile of the distribution in Fig. 5. The retrieved ice fraction is interpreted as the mean ice fraction of the layer from cloud top to the optical thickness threshold. For the cloudbow range and the slope range an optical thickness threshold of about 0.8 and 1.0 was obtained for the considered case.

In addition to the two-layer cloud, a plane-parallel cloud with a linear profile of the ice fraction from 0 at cloud top to 1 at cloud base (see Fig. 4b) was simulated and the phase partitioning retrieval applied. From the derived ice fraction, the optical thickness threshold was computed such that the average ice fraction from cloud top to the threshold was equal to the retrieved ice fraction. The obtained threshold values are denoted by the dashed lines in Fig. 6.

As for all retrievals based on passive remote sensing, the vertical optical thickness to which the retrieval is sensitive to depends on the solar zenith angle. To quantify this dependence for the phase retrieval, the simulations of the two-layer cloud and the profile cloud were performed for different solar zenith angles and the vertical optical thickness threshold was determined

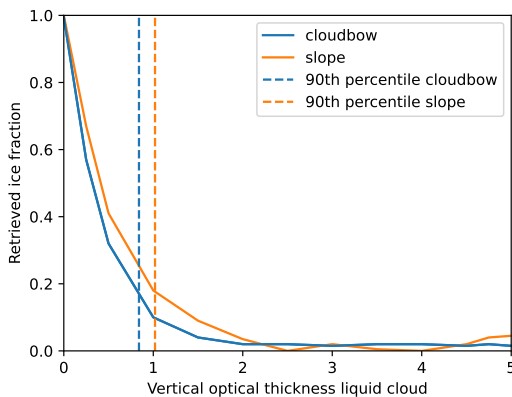

**Figure 5.** Retrieved ice fractions of the green color channel for the slope and cloudbow range for a two-layer cloud with a liquid cloud with varying optical thickness above an ice cloud, assuming a saturated polarization signal. The total optical thickness was 5, the solar zenith angle 75°. The dashed lines denote the 90th percentile of the distribution used as vertical optical thickness threshold for the cloudbow and slope range.

for all cases. Figure 6 shows the obtained optical thickness thresholds for the slope and cloudbow range obtained from the two-layer cloud and the profile cloud as discussed above, together with their mean as a function of solar zenith angle. Panel (a) shows the results under the assumption that the polarization signal is saturated and panel (b) shows the unsaturated case. For smaller solar zenith angles, the signal is sensitive to a larger optical thickness relative to the cloud top and thus to a deeper cloud layer. For the cloudbow, the vertical optical thickness threshold varies between about 2 for 30° solar zenith angle to about 1 for 75° solar zenith angle for both the saturated and unsaturated cases. In addition, the threshold values determined for the two-layer cloud and the profile cloud show only small differences between about 0 and 0.5 for the cloudbow angular range in both cases. In contrast, the differences between the two are much larger for the slope angular range. The mean varies in a similar range compared to the cloudbow range for the saturated assumption. However, the vertical optical thickness threshold for the slope range is increased if the retrieval assuming an unsaturated polarization signal is applied. This can be expected as in the unsaturated case, the ice fraction is retrieved in a combined retrieval using $I$ and $Q$. Since the cloudbow is dominated by the first scattering order, the penetration depth of $I$ and $Q$ is almost similar. For the slope range, the influence of multiple scattering is much larger and the penetration depth for $I$ increased. Consequently, the vertical optical thickness threshold of the slope range is higher for an unsaturated polarization signal.

To conclude, we define the retrieved ice fraction as the average ice fraction from cloud top to an optical thickness threshold between about 1 and 2, depending on the solar zenith angle. For 75° solar zenith angle, the optical thickness threshold is about 0.8 for the cloudbow, 1.0 for the slope with a saturated polarization signal and 1.5 for the slope in the unsaturated case. These values will be used for the validation of the retrieval with synthetic data in Sect. 5.3.

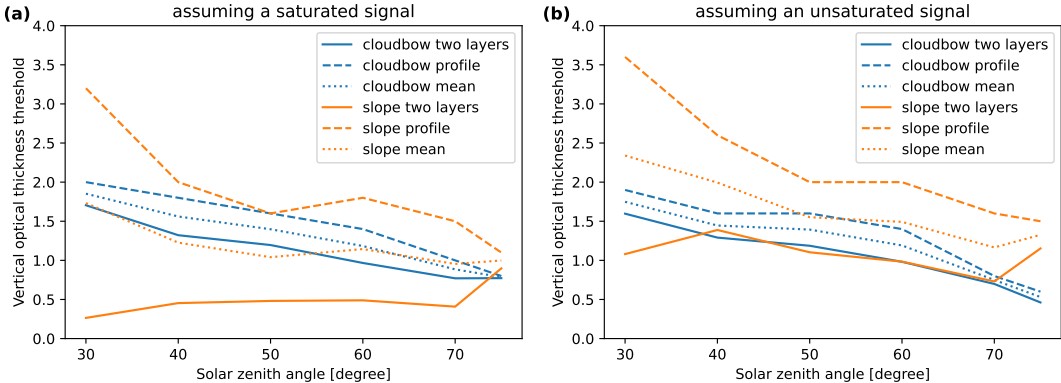

**Figure 6.** Vertical optical thickness thresholds obtained from the two-layer cloud and the profile cloud and their mean for different solar zenith angles. (a) Results for the assumption of a saturated polarization signal. (b) Results for the assumption of an unsaturated polarization signal.

## 5 Retrieval validation using synthetic data

To validate the presented retrieval and characterize its uncertainties, the retrieval was applied to synthetic data. We distinguish the different retrieval configurations: the retrieval for a saturated polarization signal using only $Q$ and the combined retrieval using $I$ and $Q$ for an unsaturated polarization signal, as well as the retrieval using the cloudbow angular range and the slope angular range. In addition, we explore differences between the assumption of plane-parallel cloud in the forward operator and the application of the IDEFAX to approximate 3D cloud geometry. The synthetic data was computed for clouds with varying complexity, for simple plane-parallel homogeneously mixed clouds, homogeneously mixed 3D clouds, and realistic 3D clouds. This allows to isolate the different contributions to the overall uncertainty, such as e.g. the effect of 3D cloud geometry.

### 5.1 Homogeneously mixed plane-parallel clouds

To begin with, the case of a homogeneously mixed plane-parallel cloud is analyzed. Such a cloud has a constant ice fraction throughout the cloud. Polarized radiative transfer simulations were performed with MYSTIC for different ice fractions and the retrieval using the forward operator for plane-parallel clouds was applied to the data for both the cloudbow and the slope angular range. In addition, we distinguish the retrieval for saturated and unsaturated polarization signals. Table 1 summarizes the results for a cloud extending from 2 to 3 km height with a total optical thickness of 5, and a solar zenith angle of 70° assuming a saturated polarization signal for three different ice fractions. In addition, the results for the combined optimization of $I$ and $Q$ assuming an unsaturated signal are given in Table 2 for plane-parallel clouds with an optical thickness of 1, 2, and 5.

Generally, there is good agreement between the true simulated ice fractions and the retrieved values, with maximum deviations of the retrieved ice fractions of 0.03. In addition, the optical thickness is retrieved correctly. This validates the retrieval method itself. The small differences between the retrieval results assuming a saturated or unsaturated polarization signal are

**Table 1.** True and retrieved ice fraction for a plane-parallel homogeneously mixed cloud assuming a saturated polarization signal (using only $Q$ for the optimization) for the green color channel of the polarization-resolving cameras. The cloud was located between 2 and 3 km height, had a total optical thickness of 5 and the solar zenith angle was $70°$. Liquid and ice effective radius were $10\,\mu m$ and $50\,\mu m$, respectively.

| | | | |
|---|---|---|---|
| True ice fraction | 0.1 | 0.5 | 0.9 |
| Retrieved ice fraction cloudbow | 0.11 | 0.51 | 0.90 |
| Retrieved ice fraction slope | 0.13 | 0.51 | 0.92 |

**Table 2.** True and retrieved ice fraction and optical thickness for a plane-parallel homogeneously mixed cloud assuming an unsaturated polarization signal (using $I$ and $Q$ combined for the optimization) for the green color channel of the polarization-resolving cameras. The cloud was located between 2 and 3 km height, had a total optical thickness of 1, 2, and 5, respectively, and the solar zenith angle was $70°$. Liquid and ice effective radius were $10\,\mu m$ and $50\,\mu m$.

| | | | | | | | | | |
|---|---|---|---|---|---|---|---|---|---|
| True ice fraction | 0.1 | 0.5 | 0.9 | 0.1 | 0.5 | 0.9 | 0.1 | 0.5 | 0.9 |
| Retrieved ice fraction cloudbow | 0.10 | 0.50 | 0.90 | 0.10 | 0.50 | 0.90 | 0.10 | 0.52 | 0.90 |
| Retrieved ice fraction slope | 0.08 | 0.50 | 0.90 | 0.08 | 0.48 | 0.90 | 0.10 | 0.50 | 0.90 |
| True optical thickness | 1.0 | 1.0 | 1.0 | 2.0 | 2.0 | 2.0 | 5.0 | 5.0 | 5.0 |
| Retrieved optical thickness cloudbow | 1.0 | 1.0 | 1.0 | 2.0 | 2.0 | 2.0 | 5.0 | 5.0 | 5.0 |
| Retrieved optical thickness slope | 1.0 | 1.0 | 1.0 | 2.0 | 2.0 | 2.0 | 5.0 | 5.0 | 5.0 |

due to the different optimization methods in both cases. The difference between the different retrieved and true values also includes uncertainties of the forward operator, which are obviously small for plane-parallel clouds.

## 5.2 Homogeneously mixed 3D clouds

Realistic clouds generally have a 3D cloud geometry that differs from the plane-parallel cloud assumption. Thus, the retrieval errors introduced by 3D cloud geometry are analyzed next. Here, the clouds are still assumed to be homogeneously mixed with a spatially uniform ice fraction to focus on the geometry only. As before, we distinguish between the cloudbow and the slope angular range and discuss differences between the retrievals for saturated and unsaturated polarization signals. In addition, we now apply the retrieval using the plane-parallel approximation in the forward operator and using the IDEFAX and compare the results.

A cloud field with a realistic cloud geometry was simulated with the Weather Research and Forecasting (WRF) model (Skamarock et al., 2019) as described in Weber et al. (2025). The WRF simulations were forced with ERA5 data for 2022-04-01 and performed with $100\,m$ horizontal resolution. More details to the WRF simulations can be found in Weber et al. (2025). The obtained cloud field was then used as input for polarized 3D radiative transfer simulations to compute synthetic

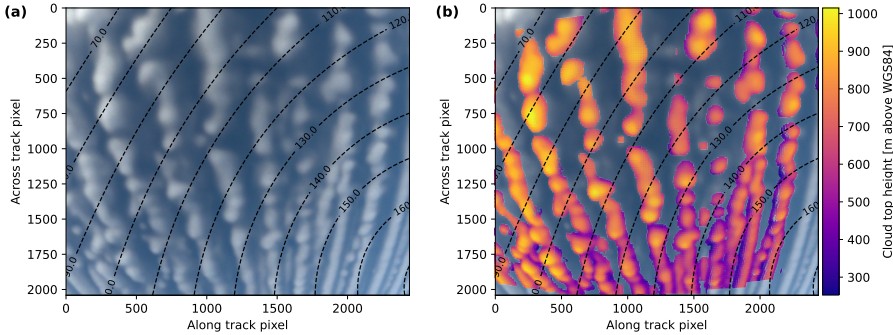

**Figure 7.** Simulation of the polLR camera of specMACS with the same viewing geometry as the measurements on 2022-04-01 at 10:18 UTC. (a) RGB image. (b) Retrieved cloud top height for all cloud targets. The dashed lines indicate the scattering angles.

observations of the specMACS polarization-resolving cameras. The radiative transfer simulations were performed with the Monte Carlo radiative transfer solver MYSTIC. The synthetic measurement data were computed as described in Volkmer et al. (2024). However, the simulations in this work did not include aerosol and we used the cloud field for low-level Arctic mixed-phase clouds from WRF instead of tropical shallow cumulus clouds. The simulated viewing geometries correspond to measurements during the HALO–$(\mathcal{AC})^3$ campaign on 2022-04-01 between 10:17:15 and 10:18:45 UTC with 1 Hz temporal resolution. In this time range, the solar zenith angle was $75.6°$. Furthermore, we simulated spectra between 380 nm and 690 nm with 10 nm resolution, which we integrated to the red, green, and blue color channels using the spectral response functions from Weber et al. (2024). The optical properties of liquid water clouds are calculated using Mie theory (Mie, 1908) and the optical properties by Yang et al. (2013) for the aggregate of eight columns are used for ice clouds. The standard deviation of the Monte Carlo simulations was about 6 %, similar to Volkmer et al. (2024). Further explanations to the settings for the radiative transfer simulations to simulate realistic specMACS measurements are given in Volkmer et al. (2024).

Figure 7 shows the RGB image and retrieved cloud top heights for the synthetic measurement data. The viewing geometry corresponds to the measurements at 10:18 UTC shown also in Fig. 12. On 2022-04-01 a cold air outbreak was observed in the Fram Strait. During cold air outbreaks, cold and dry polar air masses are advected off the sea ice edge across comparably warm open ocean (Fletcher et al., 2016; Papritz and Spengler, 2017). The clouds formed in the initial phase of the cold air outbreak typically organize into cloud streets, which are also visible in Fig. 7. The average cloud top height is about 675 m. A comparison of Fig. 7 and 12 indicates that the obtained synthetic observations and thus the cloud field simulated with the WRF model are realistic. However, the modeled clouds are smoother compared to the observed ones, which is caused by the lower spatial resolution of the WRF simulations due to the high computational cost. In addition, Arteaga et al. (2024) compared WRF simulations of Arctic mixed-phase clouds to in situ and radar measurements and found discrepancies between model and observations such as an underestimation of ice crystal number concentrations by the model, but the WRF simulations were in general able to reproduce the overall structure of the clouds and the measured radar signals showed good agreement to the simulated signals. Thus, the synthetic measurements are suitable for a validation of the phase partitioning retrieval.

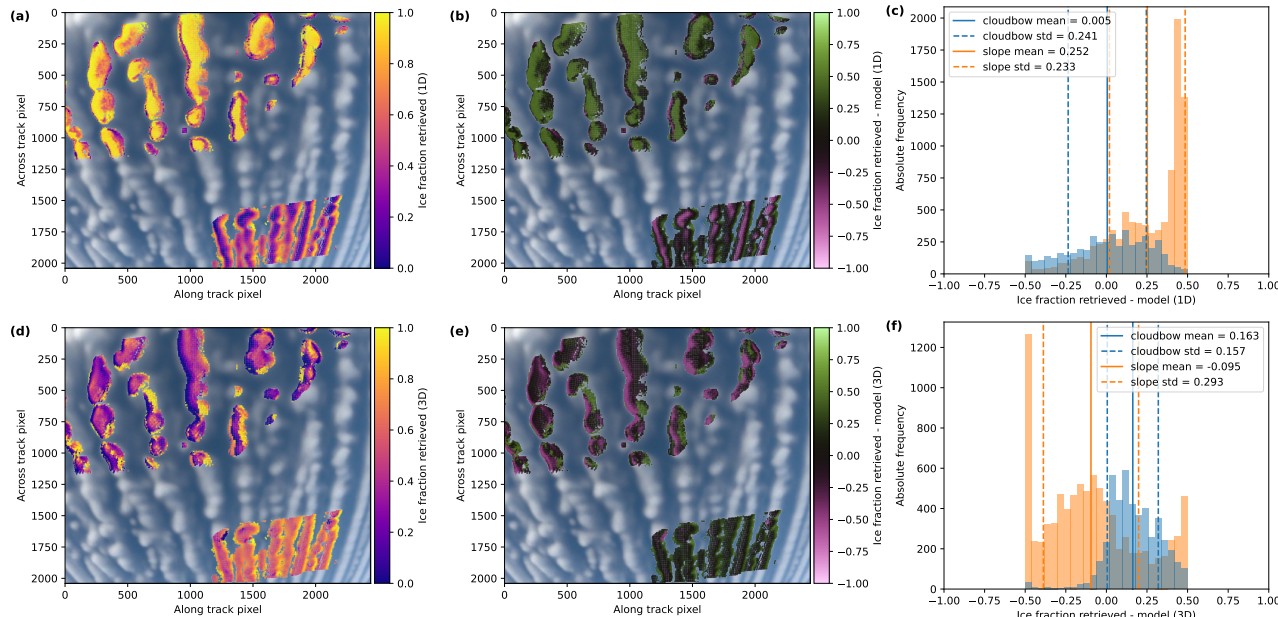

**Figure 8.** Retrieval results for the green color channel of the synthetic data for homogeneously mixed 3D clouds for the polLR camera of specMACS. (a, d) Retrieved ice fraction. (b, e) Difference between the retrieved ice fraction and the model ice fraction. (c, f) Histogram of the differences between retrieved and model ice fractions with mean and standard deviation calculated from analysis performed in the cloudbow scattering angle range (blue) and the slope range (orange). The upper row (a, b, c) shows the results under the assumption of 1D clouds, the lower row (d, e, f) shows the results using the IDEFAX. The retrieved values in the upper left part of the images correspond to the slope range and the ones in the lower right part to the cloudbow range.

The cloud field obtained from the WRF simulations has a spatially non-uniform distribution of liquid water and ice. To study the isolated effect of the 3D cloud geometry on the retrieval results, the cloud field was converted to a field of homogeneously mixed clouds with a constant ice fraction of 0.5, keeping the total optical thickness and thus the 3D cloud geometry conserved. Then, synthetic measurements were computed through 3D radiative transfer simulations with MYSTIC as described above. Afterwards, L1C data were computed from the synthetic measurement data and the cloudbow and phase retrievals were applied as described in Sect. 3.

Figures 8 and 9 show retrieval results for the forward operators using the plane-parallel independent column approximation in the upper row and the IDEFAX in the lower row. Panels (a) and (d) display the retrieved ice fractions and optical thickness values, panels (b) and (e) the difference between retrieved and true values, and panels (c) and (f) histograms of the differences, including mean and standard deviation for the cloudbow and the slope range in blue and orange, respectively. The retrieved values in the lower right of panels (a,b,d,e) correspond to the cloudbow range and the values in the upper left part to the slope range. Retrieval results are only available for parts of the images since only a 90 s period was simulated. The true ice fraction was 0.5 throughout the cloud field. For the optical thickness, the vertical optical thickness at the location of every cloud target

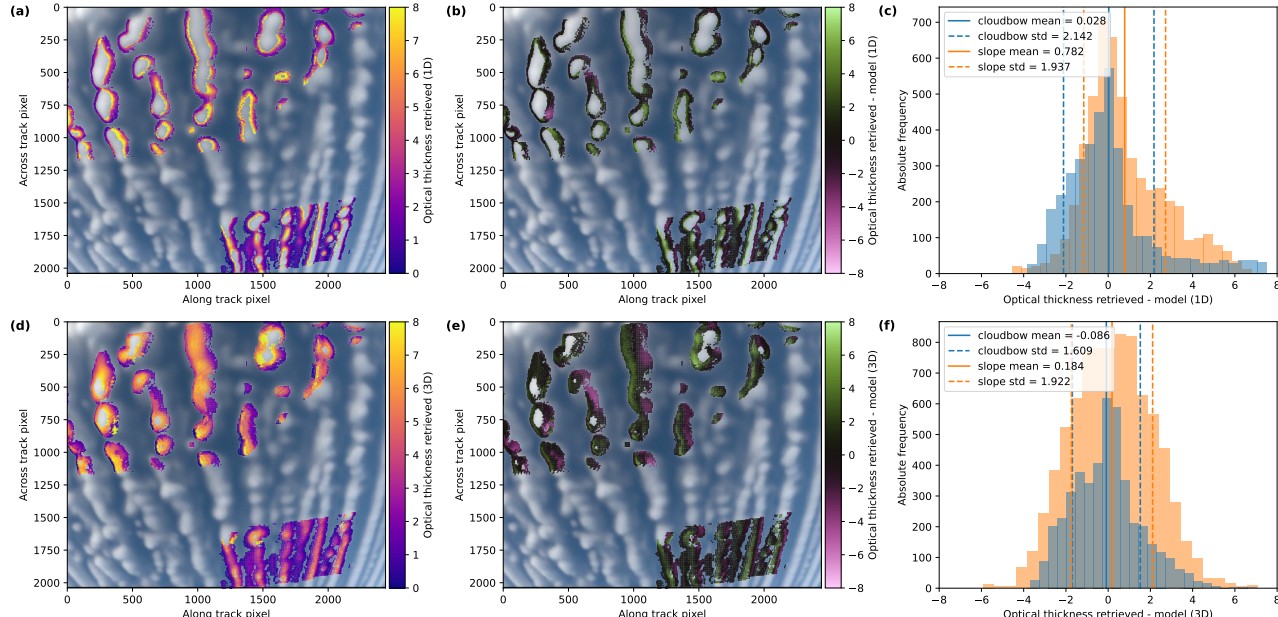

**Figure 9.** Retrieval results for the green color channel of the synthetic data for homogeneously mixed 3D clouds for the polLR camera of specMACS. (a, d) Retrieved optical thickness. (b, e) Difference between the retrieved optical thickness and the model optical thickness. (c, f) Histogram of the differences between retrieved and model optical thickness with mean and standard deviation calculated from analysis performed in the cloudbow scattering angle range (blue) and the slope range (orange). The upper row (a, b, c) shows the results under the assumption of 1D clouds, the lower row (d, e, f) shows the results using the IDEFAX. The retrieved values in the upper left part of the images correspond to the slope range and the ones in the lower right part to the cloudbow range.

point was computed from the modeled cloud field and used as ground truth since the total optical thickness is derived from the absolute intensity and represents the entire cloud column. The optical thickness is only retrieved at the cloud edges, where the clouds are optically thinner and the polarization signal is not saturated. Hence, pixels showing both a retrieved ice fraction and optical thickness in Fig. 8 and 9 have an unsaturated polarization signal and undergone the combined retrieval using $I$ and $Q$. Pixel with a retrieved ice fraction and without a retrieved optical thickness correspond to pixels with a saturated polarization signal for which the ice fraction is directly derived from $Q$. In the future, the forward operator could be extended to larger optical thicknesses to derive the optical thickness for the entire clouds after the ice fraction is known. Due to the focus on phase partitioning and the significant increase in computation time for radiative transfer simulations of optically thicker clouds, the optical thickness range was restricted to smaller values in this work.

For both plane-parallel clouds and the IDEFAX, the performance of the ice fraction retrieval is different for the cloudbow range (lower right part of the images, blue histogram) and the slope range (upper left part of the images, orange histogram) in Fig. 8. The cloudbow range shows a small mean bias even for the plane-parallel assumption, with a mean difference between the retrieved and true ice fraction of 0.005±0.241. The application of the IDEFAX leads to a small overestimation of the ice

fraction but significantly reduces the standard deviation with a difference of 0.163±0.157. In the slope range, the ice fraction is strongly overestimated under the plane-parallel assumption with an average difference between the retrieved and true values

of 0.252±0.233. With the IDEFAX, the bias of the retrieved ice fraction is significantly reduced and the ice fraction is slightly underestimated with a difference of -0.095±0.293.

The optical thickness shown in Fig. 9 increases towards the centers of the clouds. The performance of the cloudbow and the slope range is here more similar, with slightly smaller mean biases for the cloudbow range than for the slope range. For the cloudbow range, the IDEFAX shows a slightly larger mean bias but smaller standard deviation of the differences between

the retrieved and true optical thickness with -0.086±1.609 compared to 0.028±2.142 for plane-parallel clouds. In the slope range, the IDEFAX reduces both the mean difference and the standard deviation from 0.782±1.937 for plane-parallel clouds to 0.184±1.922.

The explanation for the deviations of the retrieved values from the true ice fraction and optical thickness is the deviation of the simulated polarization signals in the plane-parallel approximation and the IDEFAX from full polarized 3D radiative

transfer simulations (see also Fig. 6 in Weber et al. (2025)). In addition, the larger area being classified as unsaturated (and thus the larger area with retrieved optical thickness values) for the IDEFAX is due to the differences between the simulated total reflectivity, which were used for the classification as described in Sect. 3. For the plane-parallel approximation, the differences between retrieved and true ice fraction are especially large in the centers of the clouds, whereas the cloud sides dominate the error for the IDEFAX. The improvement of the results for the IDEFAX compared to the plane-parallel assumption is larger

for the slope range than for the cloudbow range. This can be expected since the cloudbow is more strongly dominated by the first scattering order than the slope range and thus less affected by 3D radiative effects. As shown by Weber et al. (2025), 3D radiative effects can efficiently be approximated by the IDEFAX with half-spherical clouds for this type of clouds. However, the remaining 3D radiative effects still introduce retrieval errors as the standard deviations for the ice fraction and optical thickness are still large and the mean biases are not negligible even for the IDEFAX. In addition, the uncertainty of the forward

operator using the IDEFAX is slightly larger than the uncertainty of the plane-parallel forward operator (Weber et al., 2025).

In principle, the retrieval errors due to 3D cloud geometry could be accounted for in the future, as 3D cloud geometry is known from the stereographic retrieval. Currently, full 3D radiative transfer simulations are too expensive to be applied in the retrieval, but a fast (potentially machine-learning-based) method for full 3D radiative transfer simulations might become available in the future. Then, full 3D radiative transfer simulations could be performed during the optimization and the retrieval

could be adapted and extended accordingly.

### 5.3 Realistic 3D clouds

The analysis in the previous section considered clouds with a realistic cloud geometry but with a spatially uniform ice fraction and thus homogeneously mixed clouds. In reality, cloud thermodynamic phase partitioning is not constant throughout a cloud. In low-level Arctic mixed-phase clouds, a geometrically thin liquid layer is usually located at the cloud top and ice can be

found below. As discussed in Sect. 4, a vertical profile of the ice fraction has to be assumed in the retrieval. We defined the retrieved ice fraction as the effective ice fraction of the uppermost cloud layer to a certain optical thickness threshold under the

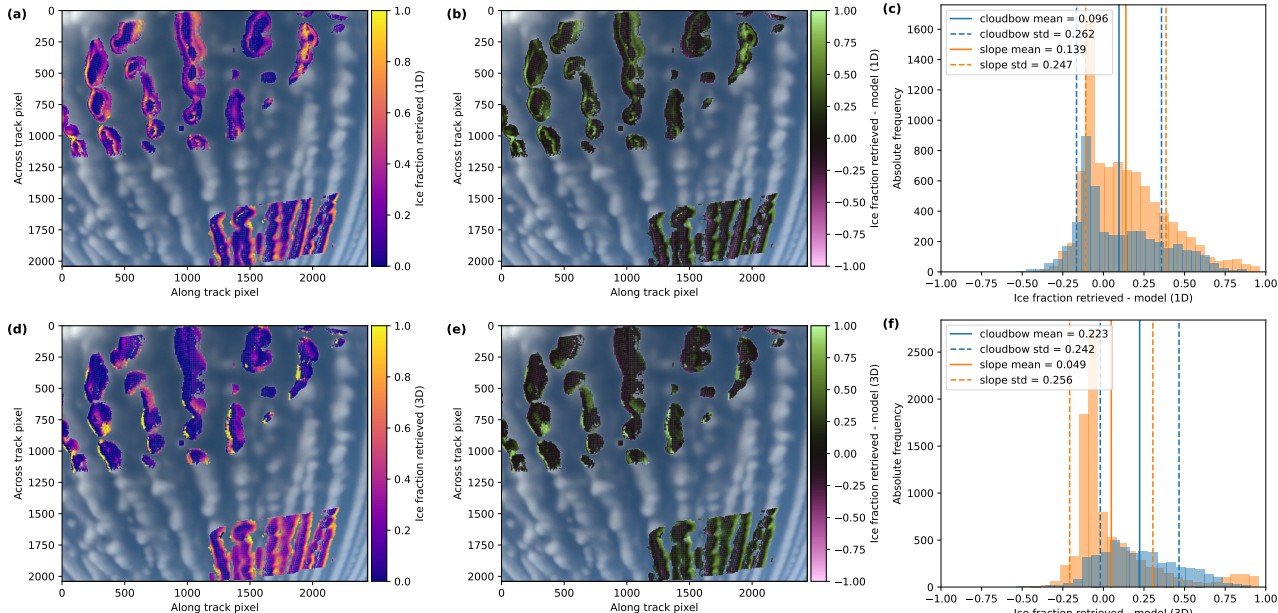

**Figure 10.** Retrieval results for the green color channel of the synthetic data for realistic 3D clouds for the polLR camera of specMACS. (a, d) Retrieved ice fraction. (b, e) Difference between the retrieved ice fraction and the model ice fraction. (c, f) Histogram of the differences between retrieved and model ice fractions with mean and standard deviation calculated from analysis performed in the cloudbow scattering angle range (blue) and the slope range (orange). The upper row (a, b, c) shows the results under the assumption of plane-parallel clouds by the retrieval, the lower row (d, e, f) the results using the IDEFAX. The retrieved values in the upper left part of the images correspond to the slope range and the ones in the lower right part to the cloudbow range.

assumption of a homogeneously mixed cloud. In the following, the agreement of the retrieval results with this assumption is tested for realistic clouds and the uncertainty introduced by the necessary assumption of a vertical ice fraction profile is studied by applying the phase retrieval to synthetic measurement data simulated with the original, realistic cloud field obtained from the WRF simulations with spatially non-uniform phase partitioning. The retrieval errors then include both errors due to 3D cloud geometry and due to the vertical ice fraction profile. As before, we distinguish the retrievals for the cloudbow and slope angular range and the retrievals using the plane-parallel approximation and the IDEFAX. In addition, we discuss differences between the retrievals for saturated and unsaturated polarization signals.

Figures 10 and 11 show the retrieval results assuming plane-parallel clouds in the upper row and using the IDEFAX in the lower row. Panels (a) and (d) display the retrieved ice fractions and optical thickness values, (b) and (e) the difference of the retrieved values to the ground truth from the model, and (c) and (f) histograms of the differences, including mean and standard deviation. To determine the ground truth for the ice fraction, optical thickness thresholds of 0.8 for the cloudbow range, 1.0 for the slope range with a saturated polarization signal, and 1.5 for the slope range in the unsaturated case, as described in Sect. 4,



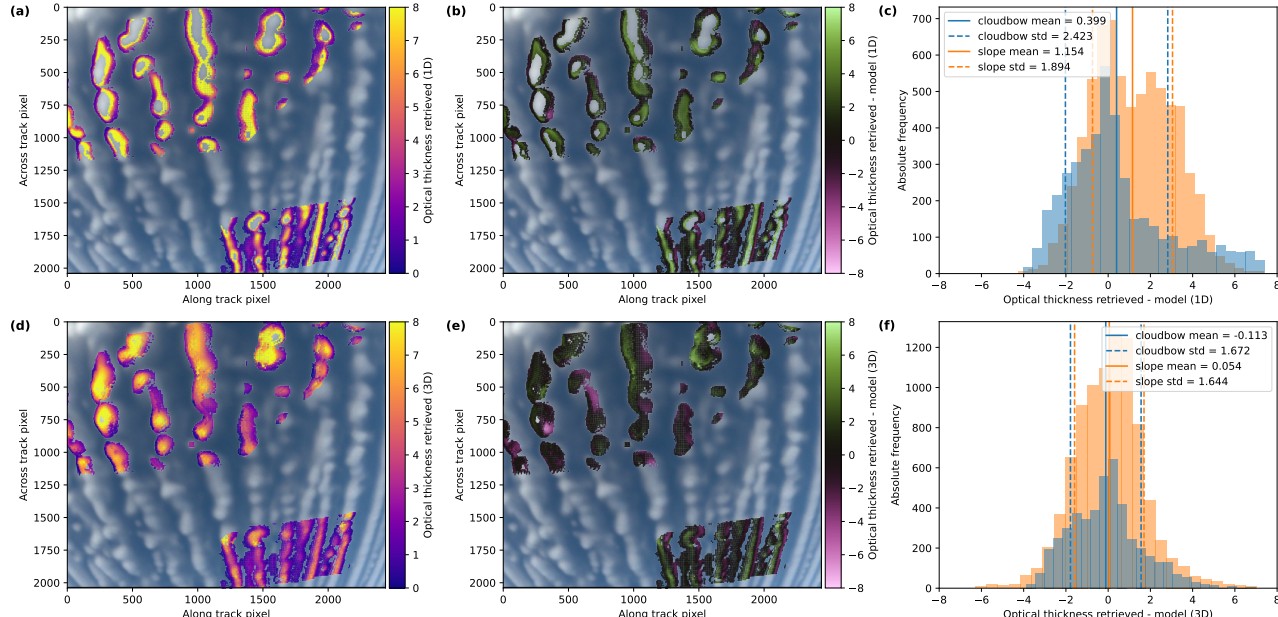

**Figure 11.** Retrieval results for the green color channel of the synthetic data for realistic 3D clouds for the polLR camera of specMACS. (a, d) Retrieved optical thickness. (b, e) Difference between the retrieved optical thickness and the model optical thickness. (c, f) Histogram of the differences between retrieved and model optical thicknesses with mean and standard deviation calculated from analysis performed in the cloudbow scattering angle range (blue) and the slope range (orange). The upper row (a, b, c) shows the results under the assumption of plane-parallel clouds by the retrieval, the lower row (d, e, f) the results using the IDEFAX. The retrieved values in the upper left part of the images correspond to the slope range and the ones in the lower right part to the cloudbow range.

were used. For the optical thickness, the column vertical optical thickness at the location of every cloud target was used as ground truth, as in Sect. 5.2. The shadow mask was applied to the results.

As before, the behavior of the cloudbow range and the slope range is different. The IDEFAX again shows a larger bias but a slightly smaller standard deviation in the cloudbow range with an average difference between retrieved and true ice fraction of $0.223\pm0.242$ compared to $0.096\pm0.262$ for the plane-parallel approximation (see Fig. 10). For the slope range, the IDEFAX reduces the mean and slightly increases the standard deviation of the differences from $0.139\pm0.247$ for plane-parallel clouds to $0.049\pm0.256$. Overall, the differences of the ice fractions are still larger towards the cloud sides compared to the cloud centers. Moreover, a step from small to large ice fractions is visible between the retrieved ice fractions for saturated and unsaturated regions in the slope range for the plane-parallel approximation in Fig. 10a. The step is due to the different penetration depths of $I$ and $Q$ and hence different signal locations for the saturated and unsaturated retrievals in this angular range as discussed in Sect. 4 (see also Fig. 6). Generally, the retrieval errors are largest for the slope angular range and the unsaturated retrieval, as the different penetration depths of $I$ and $Q$ in the combined retrieval introduce additional uncertainties. For optically thin parts of the clouds, where the optical thickness does not exceed the penetration depth of the polarization signal, the error is small

again. It is, however, increased for the intermediate range between optically thin clouds with unsaturated polarization signal and clouds with saturated polarization signal.

In comparison to the homogeneously mixed 3D clouds analyzed in Sect. 5.2, the mean differences and standard deviations are increased in the cloudbow range for the realistic 3D clouds for plane-parallel clouds and the IDEFAX. These increased errors for the realistic clouds are due to the assumption of a homogeneously mixed cloud in the forward operator and the variable vertical profile of the ice fraction in the simulated cloud field, in addition to the 3D cloud geometry. In the slope range, however, the errors of the retrieved ice fraction are slightly smaller for realistic clouds than for homogeneously mixed 3D clouds. There seems to be a compensating effect between the influence of 3D cloud geometry and the assumption of the vertical ice fraction profile, which leads to smaller mean and standard deviation values of the differences between the retrieved and true ice fraction. In all cases, however, there is a significant contribution of the 3D cloud geometry to the total retrieval error. The influence of the assumption of the vertical ice fraction profile can also not be neglected, as indicated by the differences between the error histograms in Fig. 8 and Fig. 10.

Concerning the optical thickness displayed in Fig. 11, the mean differences and standard deviations are smaller for the IDEFAX than for plane-parallel clouds and smaller for the cloudbow range than the slope range, similar to the homogeneously mixed 3D clouds discussed in Sect. 5.2. In all cases, the mean differences and standard deviations are only slightly increased compared to the homogeneously mixed clouds. This indicates that the error due to 3D cloud geometry dominates over the error due to the assumption of a vertical ice fraction profile for the optical thickness.

The differences between retrieval and model in Fig. 10 and 11 include the errors due to the assumption of a homogeneously mixed cloud, respectively a vertical ice fraction profile, as well as errors due to 3D cloud geometry. Since no information about the vertical ice fraction profile is available from passive remote sensing information, the corresponding error cannot be reduced. The error due to 3D cloud geometry, however, could be further reduced in the future if fast full 3D radiative transfer models become available. In addition, the definition of the ground truth used for the validation is not trivial. Here, the vertical mean between cloud top and the optical thickness threshold values at the location of the cloud target is used as discussed above and in Sect. 4. However, the observed deviation between the retrieved values and the so-defined model values could also be partly explained by a deviation of the real ground truth from the chosen model ground truth. The obtained errors thus have to be interpreted as retrieval uncertainties under these assumptions.

The errors of the retrieved ice fractions are not negligible. However, accurate measurements of cloud phase partitioning in mixed-phase clouds are challenging even for in situ measurements (Korolev et al., 2017). Cloud phase partitioning can quantitatively be computed from in situ measurements of liquid water content and total water content. Following the uncertainties of cloud water content measurements in liquid clouds given in Faber et al. (2018) and ice clouds specified by Heymsfield et al. (2010) and Hogan et al. (2012), Moser et al. (2023) concluded that in situ measurements of cloud water content in mixed-phase clouds have uncertainties in the range of 20 % to 50 %. The uncertainty of the new polarimetric retrieval of cloud phase partitioning is in the same range, but it provides two-dimensional fields of the retrieved quantities with high spatial resolution. In addition, it is very sensitive to liquid water at cloud top and can detect even small amounts of liquid water, which is difficult to detect with other passive remote sensing methods.

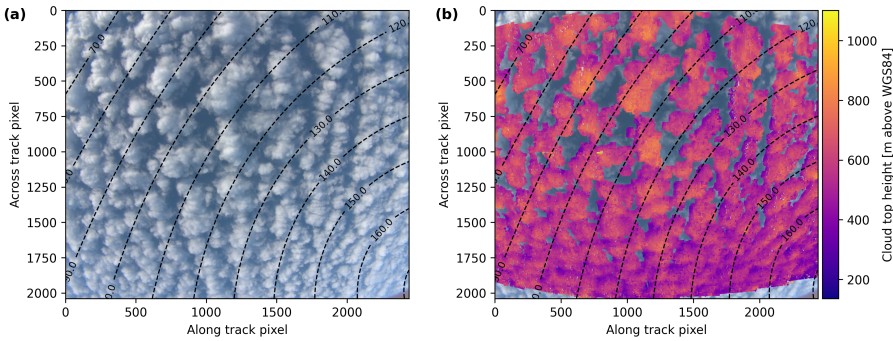

**Figure 12.** Measurement of the polLR camera of specMACS on 2022-04-01 at 10:18 UTC. (a) RGB image. (b) Cloud top height for all cloud targets. The dashed lines indicate the scattering angles.

## 6   Application to measurement data

As an example case study, the retrieval of cloud phase partitioning is finally applied to measurements of the specMACS instrument during the HALO–$(\mathcal{AC})^3$ measurement campaign (Wendisch et al., 2024; Ehrlich et al., 2025) which took place in Kiruna and Svalbard in March and April 2022 and focused on air mass transformations during meridional transports into and out of the Arctic. Figure 12 shows measurements of the polLR camera of specMACS on 2022-04-01 at 10:18 UTC. These measurements correspond to the simulated data shown in the previous section. The left panel shows the RGB image and the right panel the cloud top heights for all cloud targets. Observed cloud top heights are between 400 m and 800 m with a mean cloud top height of about 570 m and cloudbow and slope range are both covered by the measurements.

Here, only the retrieval results for the neural network forward operator using the IDEFAX are presented since the IDEFAX was shown to accurately approximate the 3D cloud geometry for the considered type of low-level Arctic mixed-phase clouds (Weber et al., 2024) and the retrieval using the IDEFAX generally demonstrated a better performance as discussed in Sect. 5. Figure 13 shows the results of the phase retrieval, before applying the shadow mask. Panels (a) and (b) display the retrieved ice fraction and optical thickness and panels (c) and (d) the root mean square errors of the fits for $I$ and $Q$. In contrast to the synthetic data, where only a short time period was simulated, the entire observation range in the along-track direction can be analyzed for the measurements. Moreover, the measurements have measurement uncertainties whose influence is included in the RMSEs in Fig. 13c and 13d. The results in the upper part of the panels correspond to the slope angular range, whereas the results in the lower part are for the cloudbow angular range. Evaluation of the area in between is not possible since neither the slope nor the cloudbow angular range are completely observed for these cloud targets. The complete angular range from 135.9° to 160° is needed to determine the cloud droplet size distribution for the phase retrieval in the cloudbow angular range. In addition, the retrieval for the slope range is only possible for minimum scattering angles of 80° and smaller since the sensitivity of the slope angular range to cloud thermodynamic phase decreases with increasing minimum scattering angle, as can be seen in Fig. 1, and the retrieved ice fractions become very uncertain for a further reduced scattering angle range.

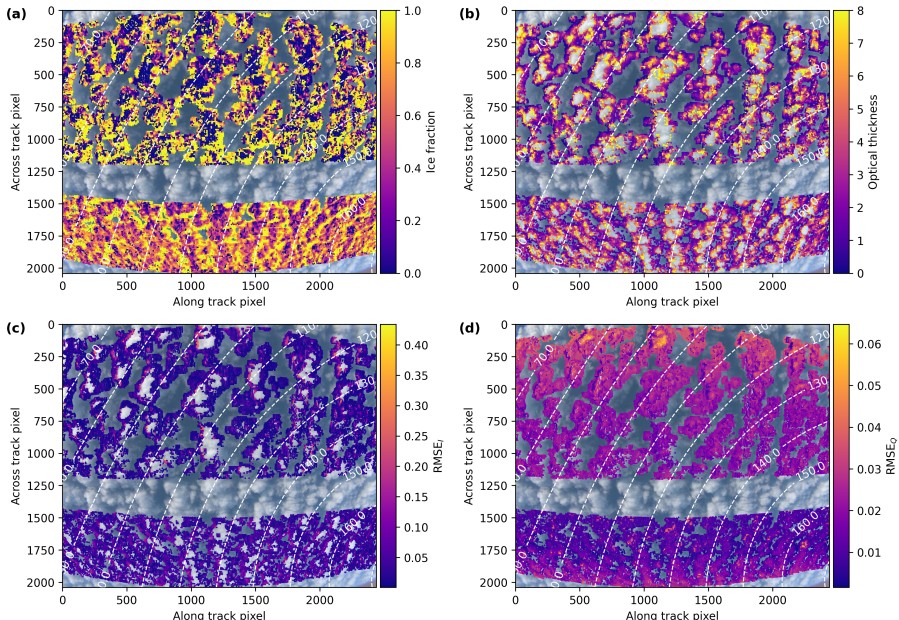

**Figure 13.** Results of the phase partitioning retrieval with the neural network forward operator for half-spherical clouds with the IDEFAX for the green color channel of the polLR camera of specMACS on 2022-04-01 at 10:18 UTC. Panels (a) and (b) display the derived ice fraction and optical thickness, respectively, and panels (c) and (d) the RMSE for $I$ and $Q$. The dashed lines indicate scattering angles.

The results show smaller ice fractions in the (higher) centers of the clouds and larger ice fractions towards the lower cloud sides. This can be expected as Arctic mixed-phase clouds usually have a liquid layer at cloud top from which ice crystals form and sediment downwards (e.g. Morrison et al., 2012). The mean derived ice fraction is about 0.62 for the cloudbow range and 0.51 for the slope range. The cloudbow range shows more continuous transitions between liquid and ice since it is more sensitive to cloud phase partitioning than the slope range, as can also be seen in Fig. 1. In addition, the slope range is more strongly affected by 3D radiative effects as discussed in Sect. 5. Thus, whenever both the cloudbow and the slope range are covered by the measurements for the same scene, the retrieval results of the cloudbow range should be preferred for further scientific analysis, keeping, however, the biases discussed in Sect. 5 in mind. The patterns of the root mean square error (RMSE) of $I$ and $Q$ and panels (c) and (d) of Fig. 13 are related to the 3D cloud geometry as can be seen by comparing the panels to Fig. 12. Larger RMSEs for $I$ are, for example, found at the cloud sides, which are oriented towards the sun, which is coming from the upper left. Moreover, the measurement uncertainties, especially for the polarization signal, are larger towards the corners of the sensor, where calibration artifacts can occur (Weber et al., 2024). As a consequence, the $RMSE_Q$ increases towards the top and the bottom of the figure.

## 7 Summary

A new retrieval of cloud thermodynamic phase partitioning from multi-angle polarimetric imaging was presented and applied to measurements of the polarization-resolving cameras of the specMACS instrument. The basic idea of the retrieval is to fit measured multi-angle signals of $Q$ along the scattering plane to simulated multi-angle signals to obtain an ice fraction, which is defined as the ratio of the ice optical thickness to the total optical thickness. In contrast to existing retrievals based on spectral measurements, the presented polarimetric retrieval provides quantitative information about cloud phase partitioning. The obtained ice fractions are representative for the cloud top as polarization is dominated by single scattering. For the same reason, polarization is less influenced by 3D radiative effects and the localization of the measured signal is more confined compared to the total reflectivity. 3D radiative effects are accounted for by using the IDEFAX introduced by Weber et al. (2025), where 3D cloud geometry is approximated through half-spherical clouds. Information about 3D cloud geometry needed for the IDEFAX can be derived from the specMACS observations by applying a stereographic retrieval (Kölling et al., 2019). Depending on the observed cloud structure, the assumption of either plane-parallel or half-spherical clouds in the IDEFAX will be more suited (Weber et al., 2025) and the retrieval can be performed using the respective forward operator for both assumptions.

Next, the retrieval was validated using synthetic data. Idealized cloud cases as well as a realistic cloud field of Arctic mixed-phase clouds simulated with the WRF model were used as input for 3D radiative transfer simulations. Then, the retrieval was applied to the synthetic data and the retrieved quantities were compared to the true quantities from the model. The good agreement between retrieved and true values for plane-parallel homogeneously mixed clouds validated the retrieval method itself. For realistic 3D Arctic mixed-phase clouds, mean differences between retrieved and true ice fractions of $0.096\pm0.262$ and $0.139\pm0.247$ were obtained for the cloudbow and the slope range, respectively, for the plane-parallel approximation. For the IDEFAX, the analysis showed mean differences of $0.223\pm0.242$ respectively $0.049\pm0.256$. These errors include errors due to 3D cloud geometry as well as errors due to the necessary assumption of a vertical ice fraction profile. To quantify the contributions of 3D cloud geometry and the assumption of a vertical ice fraction profile to the total error, additional synthetic data were computed for the realistic cloud field obtained from the WRF model, but with a spatially uniform ice fraction. Both the error due to 3D cloud geometry as well as due to the assumption of the vertical ice fraction profile have a significant contribution to the total error of the retrieved ice fraction. Generally, the retrieval using the cloudbow angular range is more accurate than the retrieval using the slope angular range, as the cloudbow range is more sensitive to the cloud thermodynamic phase and more strongly dominated by single scattering, and should therefore be preferred if the observation of the cloudbow is geometrically possible. In addition, the uncertainty of the retrieval results for saturated polarization signals is higher than for unsaturated signals. In the former case, the ice fraction can directly be derived from measurements of $Q$, whereas in the latter a combined retrieval using $I$ and $Q$ has to be applied. $I$ is more strongly affected by 3D radiative effects than $Q$, and additionally the signals of $I$ and $Q$ originate from different penetration depths within the cloud, which increases the uncertainty of the derived ice fraction.

The retrieval errors are not negligible. However, quantitative measurements of cloud thermodynamic phase partitioning in mixed-phase clouds are challenging even for in situ measurements, which also have large uncertainties in this case and only provide one-dimensional information along the flight track. Even though the retrieved ice fractions have large uncertainties, the new polarized retrieval can detect very small amounts of liquid cloud droplets in an ice cloud layer, which can not be achieved with other passive remote sensing methods. The spatial variability of the ice fraction is affected by the retrieval uncertainty due to the impact of the 3D cloud geometry and the assumption of a vertical ice fraction profile. The validation based on synthetic data, however, showed that the method works well on average for complete cloud scenes. Thus, the results can, for example, be used for a statistical analysis of the cloud thermodynamic phase partitioning of a cloud scene. Finally, the retrieval was applied to observations of low-level Arctic mixed-phase clouds during the HALO–$(\mathcal{AC})^3$ campaign and showed a liquid layer at cloud top as expected.

The presented phase retrieval and validation studies did not consider the influence of aerosol so far. Above-cloud-aerosol, in general, affects $I$ as well as $Q$ and could, for example, reduce the amplitude of the cloudbow in $Q$ (Alexandrov et al., 2012). This would in turn lead to a small overestimation of the retrieved ice fraction. The focus of this work was on measurements of mixed-phase clouds in the Arctic, where the aerosol concentrations can generally be expected to be small and the additional uncertainty introduced by aerosol is small compared to the uncertainties due to 3D cloud geometry and the assumption of a vertical ice fraction profile. In fact, satellite measurements of the aerosol optical thickness for the shown example observation on 2022-04-01 in the Fram Strait indicate small values below 0.1 for clear-sky pixels and, therefore, the influence of above-cloud-aerosol on the retrieval results was neglected. However, for other measurements in more polluted regions, the influence of above-cloud-aerosol should be considered. To this end, additional validation and sensitivity studies should be performed. Furthermore, it could be investigated if the aerosol optical thickness, obtained e.g. from satellite measurements, could be included as an additional parameter into the retrieval.

In addition, there are some further potential improvements and retrieval extensions as well as future applications of the presented retrieval. The introduced polarimetric retrieval method could be further improved by combining it with a spectral retrieval, which is sensitive to small amounts of ice in a liquid cloud, similarly to Riedi et al. (2010). This combination allows for a very accurate phase classification. In addition, retrieval results with a large RMSE of the fit can, for example, be filtered out for further analysis to increase the accuracy, since a large RMSE indicates a larger uncertainty of the retrieved ice fraction. In the future, the retrieval could be extended to derive the total optical thickness as an additional variable for all observations. The retrieval results obtained from the specMACS instrument have a high spatial resolution of about $100\,\mathrm{m}$, which allows for studying small-scale processes and the horizontal distribution of cloud thermodynamic phase. Typical scales of inhomogeneities in Arctic clouds were analyzed by Schäfer et al. (2017) and Schäfer et al. (2018) and are on the order of a few hundred meters, which is resolved by the spatial resolution of the specMACS measurements. However, the influence of the retrieval uncertainties on the small-scale variability has to be considered. In addition, the observations of cloud thermodynamic phase partitioning could be used to constrain models and potentially improve the representation of Arctic mixed-phase clouds in climate and general circulation models. The retrieval is, of course, also applicable to data measured outside the Arctic to study, for example, glaciation in deep convective clouds in the tropics. Due to the lower solar zenith angles and consequently reduced

influence of 3D radiative effects, the retrieval is expected to perform better in this situation. In addition, the more sensitive cloudbow region will then be observed closer to the center of the field of view of the instrument, where the measurement uncertainties are smaller. On the other hand, there is likely a larger influence of above-cloud-aerosol for measurements in the tropics compared to the Arctic due to the expected larger aerosol amount.

*Code and data availability.* The measurements and software code for the retrieval can be provided upon request from the corresponding author. The WRF simulations and the forward operator can be obtained from Weber et al. (2025).

*Author contributions.* CE had the idea for the retrieval. VP developed and implemented the method to compute L1C data. AW developed and implemented the phase retrieval in close discussion with BM, performed the simulations, applied the retrieval to measured and simulated data, and wrote the manuscript with input from all co-authors. VP, CE, and BM provided valuable feedback during the development of the retrieval and interpretation of the results.

*Competing interests.* Bernhard Mayer is member of the editorial board of AMT.

*Acknowledgements.* We would like to thank Gregor Köcher for performing the WRF simulations and Lea Volkmer for applying the stereographic retrieval to the specMACS osbervations from HALO–$(\mathcal{AC})^3$. This research was supported by the German Research Foundation (DFG) within the project SPP 1294 under project number 442667104.

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
