# Peer review of "Retrieval of cloud thermodynamic phase partitioning from multi-angle polarimetric imaging of Arctic mixed-phase clouds"

_EGUsphere, 2025_

## Referee Comment (RC1)

Referee comment on "Retrieval of cloud thermodynamic phase partitioning from multi-angle polarimetric imaging of Arctic mixed-phase clouds" by Weber et al.

This study presents a novel retrieval method based on multi-angle polarimetry to derive the ice fraction near the cloud top. The method involves simulating polarized signals within the cloudbow and slope ranges of the scattering angle. The retrieval was applied to synthetic cloud scenes representing Arctic mixed-phase clouds, as well as to measurements taken during the HALO-(AC)3 campaign. Uncertainties in the retrieval were attributed to 3D radiative effects and the assumption of a vertical ice fraction profile.

Overall, the authors did a very good job of presenting their new approach. The manuscript is well structured. Some suggestions for improving the content are provided below. I can recommend publication once clarification has been provided.

**General Comments**

- 1. Retrieval results for the horizontal distribution of the thermodynamic phase of clouds near the cloud tops have shown systematic features that do not necessarily provide a consistent picture of slope and cloud bow range (see Figures 10a, 10b and 13a). In any case, the spatial variability of the ice fraction is significantly impacted by retrieval uncertainty, both quantitatively and qualitatively. Would it be more accurate to claim that the method works well on average for complete cloud scenes rather than for individual cloud elements?
- 2. The authors have shown the I and Q components of the Stokes vector for 550 nm to illustrate the sensitivity to the ice fraction. The polarization camera moreover gives I and Q for a broader spectral range (R, G, B channels). How are the spectral signatures affected by this fact? Is there a lower sensitivity when accounting for the spectral resolution of the camera? Which of the three spectral bands (R, G or B) is finally used for retrieval? From the different figures, I guess it is the green channel. Perhaps I missed a discussion about which of these channels is most appropriate.
- 3. Why is the shadow masking performed after the retrieval of the ice fraction? Would it not be more efficient to mask this data in advance?

**Minor/Specific Comments**

- 1. The method is sensitive to the penetration depth of the radiation into the cloud. The retrieved ice fraction corresponds to the upper most cloud layer (optical thickness between 1 and 2). What are realistic values of the optical depth of the liquid layer at cloud top? Could you be more specific in the introduction about what mixed-phase clouds typically look like in the Arctic?
- 2. P3l82: "10 m resolution" → Specify a flight altitude that complies with this resolution.
- 3. P4 Fig1a: The figure is not discussed in this section, only Fig. 1b.
- 4. P5l121: "Here, cases where the polarization signal is saturated and not saturated are distinguished." → "Here, cases where the polarization signal of Q is saturated and not saturated are distinguished (see Sec. 3.4)." I found it difficult to understand the meaning of 'saturated' here. This becomes clear in Section 3.4.
- 5. P6l146: "... from the cloudbow range to cover the complete scattering angle range" Maybe add "available" before "scattering angle"
- 6. P6l160: "see Fig. 3b" I would suggest referring to Fig. 1b instead, since Fig. 3b has not yet been introduced.
- 7. P7l163: "In case the geometry does not allow for observing the cloudbow..." This case is not included in Fig. 2. Maybe revise the schematics.
- 8. P7l175: "but the accuracy of the retrieved ice fraction is not affected" Why not?

- 9. P7l192: "Unknown parameters are the total optical thickness" → "Unknown parameters are the total cloud optical thickness"
- 10. P8I204: "Thus, the retrieved ice fraction has to be interpreted as an effective ice fraction under the assumption of a homogeneously mixed cloud." Later, the ice fraction is related to the cloud top layer. This statement may give the reader the wrong impression, of what the effective ice fraction provides. It's not representing the whole cloud layer.
- 11. P8l214: "total intensity I are compared to simulations with a worst-case assumption of fice = 1, since ice clouds are brighter than liquid clouds" I wouldn't call it a worst-case assumption. It's rather an extreme case. Further, add "within this range of scattering angles". Ice clouds are not generally brighter than liquid clouds.
- 12. P8l221: "I and Q are converted to reflectivity as in Weber et al. (2025)" Do the previous plots show R\_I and R\_Q (notation from Weber et al., 2025) or I and Q? Which downward irradiance (E\_dw) is used to calculate the reflectivity? E\_0\*cos(theta) is only a rough estimate for E\_dw at flight altitude.
- 13. P10l249: "there are two possible extreme cases" There are certainly more extreme cases. I suggest to rewrite the sentence. "For the phase partitioning in low-level Arctic mixed-phase clouds, we assume two extreme cases for the vertical profile."
- 14. P10 Fig4: I am having difficulty understanding the illustration. Assuming the black line represents the ice fraction, Fig. 4b is understandable. However, the horizontal line in Fig. 4a, which is located between the liquid and ice phases, cannot represent the ice fraction. It must be either 0 for the upper liquid part or 1 for the lower part. Please clarify.
- 15. P11I280: "In addition, there is agreement between the threshold values determined for the two-layer cloud and the profile cloud." Does this refer to Fig. 6b? I wouldn't call it an agreement, given that the threshold values derived for the two cloud profiles differ by around 0.5.
- 16. P13l313: "A realistic cloud field..." The realistic aspect is the geometry; the cloud microphysics is not.
- 17. P14l323, l326: I suggest to combine Fig. 7 and Fig. 12 here, as Fig.12 is discussed here already.
- 18. P14 Fig7: Add "Retrieved" in front of "cloud top height" in the figure caption.
- 19. P15l355: "The cloudbow range shows a small bias even for the plane-parallel assumption..." Maybe add "mean" in front of bias. Here and elsewhere.
- 20. P16l369: "reflectance"→"reflectivity"?
- 21. P17l143: "but smaller standard deviation" Actually, the numbers are almost the same.
- 22. P18I417: "In the slope range, there seems to be a compensating effect between the influence of 3D cloud geometry and the assumption of the vertical ice fraction profile, ..." I cannot follow the argument here. Compensation may only occur in the 1D case when 3D radiative effects are not considered. However, I think the authors are referring to Fig. 10 d—f and the 3D-based retrieval. Please clarify.
- 23. P18I420: "In all cases, however, there is a significant contribution of the 3D cloud geometry to the total retrieval error. The influence of the assumption of the vertical ice fraction profile can also not be neglected." It's a quite general statement here. Can you estimate which one has the bigger effect on the retrieval?
- 24. P20l455: "... IDEFAX demonstrated a better performance ..." It's not completely true for the cloudbow range.
- 25. P20l456: "before applying the shadow mask" Why does Fig. 13 not show the final result after applying the shadow mask?
- 26. P21l460: "Evaluation of the angular range between the slope range and the cloudbow range is not possible..." I'm not sure what is meant here. Is there no comparison between the retrieved ice fractions from slope range and cloudbow range possible?

**Technical Comments**

- 1. P5l116: "so-called L1C data"  $\rightarrow$  "so-called level1C (L1C) data".
- 2. P7l187: I suggest to remove "(unknown)"
- 3. P7l188: "These include ..." Perhaps consider splitting the very long sentence.
- 4. P21 Fig 13: Number of contour lines are hard to read.

---

## Referee Comment (RC2)

Retrieval of cloud thermodynamic phase partitioning from multi-angle polarimetric imaging of Arctic mixed-phase clouds Weber et al. (2025)

**Executive summary:**

This work discusses a new algorithm for ice fraction derivation from multi-angle polarimetric cloud measurements from the specMACS instrument during the HALO-AC3 campaign. The retrieval combines this data, the IDEFAX neural network forward model defined in parallel work (Weber et al. 2025), 3D Monte Carlo radiative transfer simulations from the MYSTIC routine, WRF cloud simulations, and ERA5 re-analysis. The paper uses both intensity (I) and polarized (Q) multi-angle cloud data in two regimes: "slope", or the region between 60-80 in scattering angle, and the "cloudbow" the region between 135 and 165 in scattering angle in the retrieval. The paper concludes that realistic Arctic clouds, simulated in 3D, compare best with retrieved ice fraction and cloud optical thickness (COT) over the specMACS field-of-view.

This paper is well-within the scope for AMT. It is valuable for current or upcoming polarimetric missions, such as PACE, 3MI, the polarimeter on CO2M, HACP, and the DPC/POSP series. It is also excellent that the authors are upfront about detection, modeling, and interpretation uncertainties. However, I ask for a potentially major and minor revision prior to publication.

**Potentially major revision:**

It is unclear how above-cloud-aerosol (ACA) impacts the derivation of ice fraction at cloud top. Because this retrieval relies on a fit to Q, aerosol loading may dampen the Q-signal like ice (Alexandrov et al. 2012, section 7, figure 9). To first order, aerosols will modify the depth of I as well.

Given cloud height in the Fig. 7 and 12 domains are ~1 km at most, ACA cannot be completely ruled out – though in the Arctic, AOD is likely low. However, AOD at 0.1 and lower can have an impact on I and the depth of the primary bow signal in Q, over clouds.

Therefore, retrieved ice fraction could be overestimated relative to cloud-only simulations in the presence of ACA for "saturated" pixels. "Unsaturated" pixels may be more complex. Aerosol has a darkening effect in I over clouds in the visible, which is opposite of increasing ice fraction/COT in the paper.

The interpretation of I and Q signals is important, because ice fraction here is quantitative value, not a qualitative phase index (Reidi et al. 2010, cited in-text).

The paper makes no mention of aerosol in modeling or simulation. If this has been considered, please discuss more clearly.

If not, I recommend the following:

(Most likely) Prove that the AOD in the specMACS scenes is negligible (or in other words, not a significant component of the multi-angle I or Q signals). Check the AOD from relevant satellite overpasses during HALO-AC3 or co-incident measurements from the aircraft (if those exist). If this is true (and likely is), also add discussion on how the algorithm could be adapted to address ACA impacts on ice fraction for non-clean scenes.

(Least likely) In the rare chance that AOD is not negligible, then this is a major revision. I suggest a rescope to include AOD as a retrievable parameter in the algorithm flow. To support this, show how a range of AOD impacts ice fraction retrieval with IDEFAX for f\_ice = 0.2 for unsaturated and saturated cases (since the algorithms differ). Please demonstrate with a figure.

Alexandrov, M.D., B. Cairns, C. Emde, A.S. Ackerman, and B. van Diedenhoven, 2012: Accuracy assessments of cloud droplet size retrievals from polarized reflectance measurements by the research scanning polarimeter. Remote Sens. Environ., 125, 92-111, doi:10.1016/j.rse.2012.07.012.

**Minor revision:**

I appreciate the attention to detail in the paper, though the many study configurations can be hard to follow at times. It will be more impactful to the reader if the authors simplify the discussion and more concisely explain:

- The cloud measurement scenarios: unsaturated vs. saturated
- The retrievals: Q-based vs. I and Q-based
- The cloud modeling schemes: plane-parallel vs. IDEFAX
- The cloud interpretation: 1D vs. 3D
- Add more details on IDEFAX instead of referring the reader to Weber et al. (2025), add a table on Volkner et al. (2024) inputs to MYSTIC

**In-line comments (many related to the minor revision):**

**104, 120, and elsewhere**

"Observation of the cloudbow indicates the presence of liquid water and absence of the cloudbow a pure ice cloud." (104)

"If the cloudbow is geometrically possible but not visible, 120 the cloud consists of pure ice and the ice fraction equals to 1." (120)

See major revision above - the Q signal may appear as pure ice, but contain a mix of ACA and ice (in general). This can change the interpretation of ice fraction.

**121**

What does it mean for the polarization signal to be "saturated"? As in the top of the detector dynamic range? Or does that mean that the cloud has a COT > ~3 and therefore, "infinite" to a photon? Please explain in-text here.

I realized later on this definition is on line 215 - far too late into the paper. Please bring this up to an earlier section.

**135**

How does the Kolling et al. algorithm treat cloud sides/edges? A bit more discussion about this would be great.

**159**

How robust is the minimum checking on Q to instrument measurement noise?

**176 (and following paragraph)**

I am concerned that manual cloudbow labeling does not accurately represent the true uncertainty of the cloudbow detection, and confuses the interpretation of 3D effects and other errors in the applications later in the paper.

For example, the specMACS Q uncertainty between 3.5-6% given in Weber et al. (2024) could bury weak cloudbows in noise and add error in human interpretation. This could be where the 23.4% false detection metric is coming from.

There is evidence from this and other work (van Diedenhoven et al. (2012), and unpublished from Xu et al. on PACE/HARP2) that the ice/water detection is straightforward with multi-angle polarization statistics. As noted, the high 4% false positive metrics is likely human error as well.

Instead, I recommend a more statistical approach using Qual and RMSE metrics from Portge et al. (2023) to verify the cloudbow detection. Simple thresholds on both could differentiate real cloudbows from noise or false positives. Since the cloudbow Q fit is already part of the flow, aren't these metrics part of the calculation?

It is also valuable to have an extra category "unknown" for cloudbow cases that are ambiguous. There is a precedent for "unknown" in other cloud phase indices (esp. Reidi et al. 2010) and may clarify the results that pass RMSE minimization.

Pörtge, V., Kölling, T., Weber, A., Volkmer, L., Emde, C., Zinner, T., Forster, L., and Mayer, B.: High-spatial-resolution retrieval of cloud droplet size distribution from polarized observations of the cloudbow, Atmos. Meas. Tech., 16, 645–667, https://doi.org/10.5194/amt-16-645-2023, 2023.

van Diedenhoven, B., A. M. Fridlind, A. S. Ackerman, and B. Cairns, 2012: Evaluation of Hydrometeor Phase and Ice Properties in Cloud-Resolving Model Simulations of Tropical Deep Convection Using Radiance and Polarization Measurements. J. Atmos. Sci., 69, 3290–3314, https://doi.org/10.1175/JAS-D-11-0314.1.

Weber, A., Kölling, T., Pörtge, V., Baumgartner, A., Rammeloo, C., Zinner, T., and Mayer, B.: Polarization upgrade of specMACS: calibration and characterization of the 2D RGB polarization-resolving cameras, Atmos. Meas. Tech., 17, 1419–1439, https://doi.org/10.5194/amt-17-1419-2024, 2024.

**194**

Also aerosol optical thickness (see major revision).

**214**

Add to the end "since ice clouds are brighter than liquid clouds, in our simulated cases."

**221 and elsewhere through the paper**

All mentions of "reflectivity" should be "reflectance".

**220**

Figures 1 and 3 show that the change in Q at different COD is nowhere near the same magnitude as the change in I, but ice fraction changes to Q happen almost independently to COD.

The consequence of a combined, equally weighted RMSE for I and Q in unsaturated cases is that the "winning solution" for ice fraction may overemphasize a good I comparison over Q, where the distinct information content is.

This may explain why biases in measured vs. modeled ice fraction persist in the Figure 10f histograms for in the cloudbow range retrieval - and also why the COD retrieval compares well on 11f.

I recommend considering an error-normalized metric instead, such as:

$$\chi = (1 - w_Q) \frac{I_{meas} - I_{model}}{RMSE_{I,meas-model}} + w_Q \frac{Q_{meas} - Q_{model}}{RMSE_{O,meas-model}}, \tag{1}$$

where  $w_Q$  is an empirical weight on Q. This form allows Q to directly compensate for measurement-model differences in I.  $w_Q$  may be effective at 0.5, but may need fine tuning to emphasize the independent information content in Q relative to ice fraction.

**250**

Of the two cases shown in Figure 4, neither is labeled as "homogeneously mixed". Do you mean "linearly distributed"?

**Figure 8**

The terms "cloudbow" and "slope" for the third column histograms were not immediately obvious. Please describe this more explicitly like:

"(c, f) Histogram of the differences between retrieved and model ice fractions with mean and standard deviation calculated from analysis performed in the cloudbow scattering angle range (blue) and forward scattered slope range (orange)"

And also please harmonize other figures that may have similar discussion.

**350**

I strongly suggest adding 2-panel figure that shows spatially, over the specMACS domain:

- The cloud pixels that correspond to the slope range retrieval, and which ones to the cloudbow range retrieval
- The cloud pixels that undergo the saturated retrieval (Q only) and which ones go through the unsaturated retrieval (I and Q).

I am curious if these distributions can help explain some of the spatial variation in the 3D study row of Figure 8 (d,e,f). This will also support discussion on errors (line 354 - 385).

**Summary section**

Given that the realistic 3D cloud simulations compare the best against specMACS data - of the four retrieval combinations: unsaturated slope, unsaturated cloudbow, saturated cloudbow, saturated slope - which are the most valuable and which are least effective? It is clear from Figure 13 that they may create different results and it would be excellent to summarize under what conditions they succeed and aren't as useful.

---

## Author Comment (AC1)

**Reply to referee #1**

We thank Referee #1 for reviewing the manuscript and the valuable comments and suggestions which we address below. The responses to the referee comments are given in blue italic letters.

This study presents a novel retrieval method based on multi-angle polarimetry to derive the ice fraction near the cloud top. The method involves simulating polarized signals within the cloudbow and slope ranges of the scattering angle. The retrieval was applied to synthetic cloud scenes representing Arctic mixed-phase clouds, as well as to measurements taken during the HALO-(AC)³ campaign. Uncertainties in the retrieval were attributed to 3D radiative effects and the assumption of a vertical ice fraction profile.

Overall, the authors did a very good job of presenting their new approach. The manuscript is well structured. Some suggestions for improving the content are provided below. I can recommend publication once clarification has been provided.

**General Comments**

1. Retrieval results for the horizontal distribution of the thermodynamic phase of clouds near the cloud tops have shown systematic features that do not necessarily provide a consistent picture of slope and cloud bow range (see Figures 10a, 10b and 13a). In any case, the spatial variability of the ice fraction is significantly impacted by retrieval uncertainty, both quantitatively and qualitatively. Would it be more accurate to claim that the method works well on average for complete cloud scenes rather than for individual cloud elements?

   *Thank you very much for noting that. We agree that the spatial variability of the ice fraction is Influenced by the retrieval uncertainties, that the retrieval results are best suited for statistical analyses, and that this should be mentioned. We added a discussion about it:*
   *"The spatial variability of the ice fraction is affected by the retrieval uncertainty due to the impact of the 3D cloud geometry and the assumption of a vertical ice fraction profile. The validation based on synthetic data, however, showed that the method works well on average for complete cloud scenes. Thus, the results can, for example, be used for a statistical analysis of the cloud thermodynamic phase partitioning of a cloud scene."*

2. The authors have shown the I and Q components of the Stokes vector for 550 nm to illustrate the sensitivity to the ice fraction. The polarization camera moreover gives I and Q for a broader spectral range (R, G, B channels). How are the spectral signatures affected by this fact? Is there a lower sensitivity when accounting for the spectral resolution of the camera? Which of the three spectral bands (R, G or B) is finally used for retrieval? From the different figures, I guess it is the green channel. Perhaps I missed a discussion about which of these channels is most appropriate.

   *The sensitivity was shown for 550nm, which is very close to the center wavelength of the green color channel and used here as a representative wavelength. However, the same simulations for the different color channels of specMACS look very similar. The retrieval itself can be applied to all three color channels. In Sect. 4 to 6, however, only the results for the green color channel are shown. The blue color channel is generally more influenced by Rayleigh scattering due to the smaller wavelength, which is why the green or red channel should be preferred. The green channel, however, has a better spatial resolution than the red channel due to the Bayer pattern of the polarization filters on the sensors of the polarization-resolving cameras. Therefore, we show the results of the green color channel. We added an additional discussion about that to the paper draft, e.g.:*
   *"The shown simulations are for a wavelength of 550nm, which is close to the center wavelength of the green color channel of the polarization-resolving cameras of specMACS*

*and was used here as a representative wavelength. Simulations for the broader color channels of specMACS look very similar."*

*"The retrieval can, in general, be applied to the red, green, and blue color channels of the polarization-resolving cameras of specMACS. However, in the following sections of this paper, only results for the green color channel are shown, which should be preferred due to the smaller influence of Rayleigh scattering at this wavelength range and the higher spatial resolution of the measurements for this channel (Weber et al., 2024). The results for the other channels look similar."*

3. Why is the shadow masking performed after the retrieval of the ice fraction? Would it not be more efficient to mask this data in advance?

   *Thank you very much for noting that. You are absolutely right, it would be more computationally efficient to apply the shadow mask before the phase partitioning retrieval. We applied it afterwards to be able to have a look at the retrieval results and potential effects of shadows for all data points first and then filtered the results afterwards. We added an explanation to the corresponding section noting that the shadow mask could of course also be applied before, which would speed up the computation time:*
   *"The shadow mask could also be applied before the phase partitioning retrieval to further reduce computation time."*

**Minor/Specific Comments**

1. The method is sensitive to the penetration depth of the radiation into the cloud. The retrieved ice fraction corresponds to the upper most cloud layer (optical thickness between 1 and 2). What are realistic values of the optical depth of the liquid layer at cloud top? Could you be more specific in the introduction about what mixed-phase clouds typically look like in the Arctic?

   *We added more details about low-level Arctic mixed-phase clouds to the introduction. The typical geometrical thickness of the liquid layer is on the order of 100m and typical liquid water paths in these clouds are around 100 g/m². Thus, the optical thickness of the liquid water is on the order of 10, and thus larger than the vertical optical thickness the retrieval is sensitive to.*

2. P3l82: "10 m resolution" ⮕ Specify a flight altitude that complies with this resolution.

   *The resolution of 10m is for a typical flight altitude of 10km. We added this information.*

3. P4 Fig1a: The figure is not discussed in this section, only Fig. 1b.

   *Thank you very much for noting that. Panel a is referred to but not explained in detail. For completeness, nevertheless, we would like to keep panel 1a.*

4. P5l121: "Here, cases where the polarization signal is saturated and not saturated are distinguished." → "Here, cases where the polarization signal of Q is saturated and not saturated are distinguished (see Sec. 3.4)." I found it difficult to understand the meaning of 'saturated' here. This becomes clear in Section 3.4.

   *Changed as suggested.*

5. P6l146: "… from the cloudbow range to cover the complete scattering angle range" Maybe add "available" before "scattering angle"

   *Changed as suggested.*

6. P6l160: "see Fig. 3b" I would suggest referring to Fig. 1b instead, since Fig. 3b has not yet been introduced.

   *Changed as suggested.*

7. P7l163: "In case the geometry does not allow for observing the cloudbow…" This case is not included in Fig. 2. Maybe revise the schematics.

   *We added this case to the schematics.*

8. P7l175: "but the accuracy of the retrieved ice fraction is not affected" Why not?

    *The cloudbow detection is meant as a pre-selection of the data. For signals that are classified as cloudbow, the phase partitioning retrieval is performed next and this retrieval determines the ice fraction – not the cloudbow detection. Thus, if too many signals without a cloudbow are falsely classified as cloudbow, the phase partitioning retrieval is applied to more data than necessary, which increases the computation time. But, the ice fraction is not affected. We added more detail about that.*

9. P7l192: "Unknown parameters are the total optical thickness" → "Unknown parameters are the total cloud optical thickness"

    *Changed as suggested.*

10. P8l204: "Thus, the retrieved ice fraction has to be interpreted as an effective ice fraction under the assumption of a homogeneously mixed cloud." Later, the ice fraction is related to the cloud top layer. This statement may give the reader the wrong impression, of what the effective ice fraction provides. It's not representing the whole cloud layer.

    *We changed the sentence to: "Thus, the retrieved ice fraction has to be interpreted as an effective ice fraction of the upper most part of the cloud under the assumption of a homogeneously mixed cloud." This should make clear that it is not representing the whole cloud layer.*

11. P8l214: "total intensity I are compared to simulations with a worst-case assumption of $f_{ice}$ = 1, since ice clouds are brighter than liquid clouds" I wouldn't call it a worst-case assumption. It's rather an extreme case. Further, add "within this range of scattering angles". Ice clouds are not generally brighter than liquid clouds.

    *Changed as suggested.*

12. P8l221: "I and Q are converted to reflectivity as in Weber et al. (2025)" Do the previous plots show R_I and R_Q (notation from Weber et al., 2025) or I and Q? Which downward irradiance (E_dw) is used to calculate the reflectivity? E_0*cos(theta) is only a rough estimate for E_dw at flight altitude.

    *The plots actually show I and Q as denoted in the figure labels and explained in the text. For the optimization, we convert the measurements to reflectivity using the downward irradiance at the top of the atmosphere from the ATLAS3 data with modtran version 3.5, which is included in libRadtran. It is true that this is only a very rough estimate of the reflectivity at flight altitude. However, the only reason for converting the measurements of I and Q to reflectivity is that the neural network forward operators provide reflectivity. The conversion of simulated I and Q for the forward operators was performed the exactly same way and the the conversion was only performed to obtain roughly normalized values for the neural network training. As the conversion for both, measurements and forward operators is consistent and only a multiplication with a constant factor for every data point, it does not affect the retrieval results and I and Q are in this case equivalent to R_I and R_Q. We changed the symbols in formula 2 to R_Q for consistency and added some information about the reason for the conversion.*

13. P10l249: "there are two possible extreme cases" There are certainly more extreme cases. I suggest to rewrite the sentence. "For the phase partitioning in low-level Arctic mixed-phase clouds, we assume two extreme cases for the vertical profile."

    *Changed as suggested.*

14. P10 Fig4: I am having difficulty understanding the illustration. Assuming the black line represents the ice fraction, Fig. 4b is understandable. However, the horizontal line in Fig. 4a, which is located between the liquid and ice phases, cannot represent the ice fraction. It must be either 0 for the upper liquid part or 1 for the lower part. Please clarify.

*Yes, the black line in Figure 4a and 4b represents the ice fraction at a given height. In Figure 4a the black line denotes an ice fraction of 0 at the larger heights of the upper liquid cloud (the black line overlaps there with the y-axis) and then increases in a single step at the height of the interface between the liquid water and the ice cloud to 1 at lower altitudes. So, the use of the black lines should be consistent between both panels. We added this information to the figure capture and slightly reworded the description.*

15. P11l280: "In addition, there is agreement between the threshold values determined for the two-layer cloud and the profile cloud." Does this refer to Fig. 6b? I wouldn't call it an agreement, given that the threshold values derived for the two cloud profiles differ by around 0.5.
    *This refers to Fig. 6a and 6b, but you are right that there are differences. We changed the sentence to: "In addition, the threshold values determined for the two-layer cloud and the profile cloud show only small differences between about 0 and 0.5 for the cloudbow angular range in both cases."*

16. P13l313: "A realistic cloud field…" The realistic aspect is the geometry; the cloud microphysics is not.
    *We changed the wording to "A cloud field with realistic cloud geometry".*

17. P14l323, l326: I suggest to combine Fig. 7 and Fig. 12 here, as Fig.12 is discussed here already.
    *We have also thought about combing the two figures. Fig. 12 is actually discussed together with Fig. 7 but also in Sect. 6. The question was, where does it fit better. We finally decided to put Fig. 12 into Sect. 6, since it is related to measurement data and also very helpful for the interpretation of the retrieval results in Sect. 6, without having to jump forth and back to a combined Fig. 7.*

18. P14 Fig7: Add "Retrieved" in front of "cloud top height" in the figure caption.
    *Changed as suggested.*

19. P15l355: "The cloudbow range shows a small bias even for the plane-parallel assumption…" Maybe add "mean" in front of bias. Here and elsewhere.
    *Changed as suggested.*

20. P16l369: "reflectance"→"reflectivity"?
    *Changed as suggested.*

21. P17l143: "but smaller standard deviation" Actually, the numbers are almost the same.
    *We changed the wording to "a slightly smaller standard deviation".*

22. P18l417: "In the slope range, there seems to be a compensating effect between the influence of 3D cloud geometry and the assumption of the vertical ice fraction profile, …" I cannot follow the argument here. Compensation may only occur in the 1D case when 3D radiative effects are not considered. However, I think the authors are referring to Fig. 10 d–f and the 3D-based retrieval. Please clarify.
    *The statement refers to differences between the mean differences and standard deviations shown for homogeneously mixed 3D clouds and realistic 3D clouds in Fig. 8 and 10, respectively. For the cloudbow angular range, for the IDEFAX and plane-parallel clouds, the errors are increased for realistic clouds. The additional uncertainty due to the vertical ice fraction profile increases the error. In contrast, the errors are slightly decreased in the slope angular range. This is due to the combined influence of 3D cloud geometry and the vertical ice fraction profiles. Therefore, we wrote that the influences of both effects seem to compensate each other. We rewrote the corresponding section to make this clearer.*

23. P18l420: "In all cases, however, there is a significant contribution of the 3D cloud geometry to the total retrieval error. The influence of the assumption of the vertical ice fraction profile can also not be neglected." It's a quite general statement here. Can you estimate which one

has the bigger effect on the retrieval?

*We tried to quantify the influence of both effects by applying the retrieval to synthetic data of homogeneously mixed 3D clouds and realistic clouds in Sect. 5.2 and 5.3 and compare the mean differences and standard deviations. From the results it is hard to draw a general conclusion, besides the fact that both effects are contributing significantly to the uncertainty. For the cloudbow angular range, the effect of 3D cloud geometry seems to introduce a larger fraction of the error, especially for the 1D forward operator. For the slope angular range, it is hard to tell which effect dominates due to the compensating effects.  We added a reference to the differences in the text.*

24. P20l455: "… IDEFAX demonstrated a better performance …" It's not completely true for the cloudbow range.

*For the cloudbow angular range for realistic clouds, the mean bias is smaller for the IDEFAX and the standard deviation is almost similar to the one for 1D clouds (see Fig. 10). For the slope angular range, however, there is a larger mean bias for the IDEFAX but the width of the distribution is smaller. We added a "generally " to the sentence to make clear that there are in general improvements but in some cases of course also slightly worse results.*

25. P20l456: "before applying the shadow mask" Why does Fig. 13 not show the final result after applying the shadow mask?

*We have applied the shadow mask to the data, but the results before applying the mask showed a better overview and were easier to understand due to the larger number of data points and smaller "gaps". This is the reason why we showed the retrieval results without the mask and provided additional information about the cloud geometry (i.e. cloud top height) in Fig. 12.*

26. P21l460: "Evaluation of the angular range between the slope range and the cloudbow range is not possible…" I'm not sure what is meant here. Is there no comparison between the retrieved ice fractions from slope range and cloudbow range possible?

*In Fig. 13, there is a horizontal stripe without any retrieval results. The cloud targets in this area were observed only for scattering angles in between the slope and the cloudbow range, and neither the former nor the latter was completely covered. Therefore, no ice fractions can be derived in this case. This is what the sentence refers to. It should explain the missing data in Fig. 13. We changed and extended the explanation to:*
*"The results in the upper part of the panels correspond to the slope angular range, whereas the results in the lower part are for the cloudbow angular range. Evaluation of the area in between is not possible since neither the slope nor the cloudbow angular range are completely observed for these cloud targets. The complete angular range from 135.9° to 160° is needed to determine the cloud droplet size distribution for the phase retrieval in the cloudbow angular range. In addition, the retrieval for the slope range is only possible for minimum scattering angles of 80° and smaller since the sensitivity of the slope angular range to cloud thermodynamic phase decreases with increasing minimum scattering angle…"*

**Technical Comments**

1. P5l116: "so-called L1C data" → "so-called level1C (L1C) data".
   *Changed as suggested.*
2. P7l187: I suggest to remove "(unknown)"
   *Changed as suggested.*
3. P7l188: "These include …" Perhaps consider splitting the very long sentence.
   *We split the sentence as suggested.*
4. P21 Fig 13: Number of contour lines are hard to read.
   *We increased the font size of the contour labels.*

---

## Author Comment (AC2)

**Reply to referee #2**

We thank Brent McBride for reviewing the manuscript and the valuable comments and suggestions which we address below. The responses to the referee comments are given in blue italic letters.

**Executive summary:**

This work discusses a new algorithm for ice fraction derivation from multi-angle polarimetric cloud measurements from the specMACS instrument during the HALO-(AC)[3] campaign. The retrieval combines this data, the IDEFAX neural network forward model defined in parallel work (Weber et al. 2025), 3D Monte Carlo radiative transfer simulations from the MYSTIC routine, WRF cloud Simulations, and ERA5 re-analysis. The paper uses both intensity (I) and polarized (Q) multi-angle cloud data in two regimes: "slope", or the region between 60-80 in scattering angle, and the "cloudbow" the region between 135 and 165 in scattering angle in the retrieval. The paper concludes that realistic Arctic clouds, simulated in 3D, compare best with retrieved ice fraction and cloud optical thickness (COT) over the specMACS field-of-view.

This paper is well-within the scope for AMT. It is valuable for current or upcoming polarimetric missions, such as PACE, 3MI, the polarimeter on CO2M, HACP, and the DPC/POSP series. It is also excellent that the authors are upfront about detection, modeling, and interpretation uncertainties. However, I ask for a potentially major and minor revision prior to publication.

**Potentially major revision:**

It is unclear how above-cloud-aerosol (ACA) impacts the derivation of ice fraction at cloud top. Because this retrieval relies on a fit to Q, aerosol loading may dampen the Q-signal like ice (Alexandrov et al. 2012, section 7, figure 9). To first order, aerosols will modify the depth of I as well.

Given cloud height in the Fig. 7 and 12 domains are ~1 km at most, ACA cannot be completely ruled out – though in the Arctic, AOD is likely low. However, AOD at 0.1 and lower can have an impact on I and the depth of the primary bow signal in Q, over clouds.

Therefore, retrieved ice fraction could be overestimated relative to cloud-only simulations in the presence of ACA for "saturated" pixels. "Unsaturated" pixels may be more complex. Aerosol has a darkening effect in I over clouds in the visible, which is opposite of increasing ice fraction/COT in the paper.

The interpretation of I and Q signals is important, because ice fraction here is quantitative value, not a qualitative phase index (Reidi et al. 2010, cited in-text). The paper makes no mention of aerosol in modeling or simulation. If this has been considered, please discuss more clearly.

If not, I recommend the following:

(Most likely) Prove that the AOD in the specMACS scenes is negligible (or in other words, not a significant component of the multi-angle I or Q signals). Check the AOD from relevant satellite overpasses during HALO-AC3 or co-incident measurements from the aircraft (if those exist). If this is true (and likely is), also add discussion on how the algorithm could be adapted to address ACA impacts on ice fraction for non-clean scenes.

(Least likely) In the rare chance that AOD is not negligible, then this is a major revision. I suggest a rescope to include AOD as a retrievable parameter in the algorithm flow. To support this, show how a range of AOD impacts ice fraction retrieval with IDEFAX for f_ice = 0.2 for unsaturated and saturated cases (since the algorithms differ). Please demonstrate with a figure.

Alexandrov, M.D., B. Cairns, C. Emde, A.S. Ackerman, and B. van Diedenhoven, 2012: Accuracy assessments of cloud droplet size retrievals from polarized reflectance measurements by the research scanning polarimeter. Remote Sens. Environ., 125, 92-111, doi:10.1016/j.rse.2012.07.012.

*Thank you very much for noting and discussing the additional influence of aerosol on the presented retrieval. Aerosol has so far no been considered, as the retrieval was developed for Arctic mixed-phase clouds where the aerosol optical thickness is typically very small, as also mentioned by you. The simulations of the synthetic data did not include aerosol. For the example observation of HALO-(AC)[3], measurements of MODIS and VIIRS indicate a small aerosol optical thickness below 0.1 for 2022-04-01 in the Fram Strait region. The satellite measurements are, however, only available for clear-sky pixels. Therefore, the aerosol optical thickness above the clouds is expected to be even smaller. Compared to the other sources of uncertainty, which we analyzed in detail (3D cloud geometry and vertical ice fraction profile), the impact of above-cloud-aerosol is likely very small. Therefore, we follow your "most likely" recommendation and added a discussion about the influence of aerosol and a potential future extension and adaption of the retrieval to consider the influence of above-cloud-aerosol to the discussion at the end, but also referred to it through the paper, when appropriate.*

*"The presented phase retrieval and validation studies did not consider the influence of aerosol so far. Above-cloud-aerosol, in general, affects I as well as Q and could, for example, reduce the amplitude of the cloudbow in Q (Alexandrov et al., 2012). This would in turn lead to a small overestimation of the retrieved ice fraction. The focus of this work was on measurements of mixed-phase clouds in the Arctic, where the aerosol concentrations can generally be expected to be small and the additional uncertainty introduced by aerosol is small compared to the uncertainties due to 3D cloud geometry and the assumption of a vertical ice fraction profile. In fact, satellite measurements of the aerosol optical thickness for the shown example observation on 2022-04-01 in the Fram Strait indicate small values below 0.1 for clear-sky pixels and, therefore, the influence of above-cloud-aerosol on the retrieval results was neglected. However, for other measurements in more polluted regions, the influence of above-cloud-aerosol should be considered. To this end, additional validation and sensitivity studies should be performed. Furthermore, it could be investigated if the aerosol optical thickness, obtained e.g. from satellite measurements, could be included as an additional parameter into the retrieval."*

*For further changes throughout the paper, please see the latexdiff.*

**Minor revision:**
I appreciate the attention to detail in the paper, though the many study configurations can be hard to follow at times. It will be more impactful to the reader if the authors simplify the discussion and more concisely explain:
-   The cloud measurement scenarios: unsaturated vs. saturated
-   The retrievals: Q-based vs. I and Q-based
-   The cloud modeling schemes: plane-parallel vs. IDEFAX
-   The cloud interpretation: 1D vs. 3D
-   Add more details on IDEFAX instead of referring the reader to Weber et al. (2025), add a table on Volkmer et al. (2024) inputs to MYSTIC

*Thank you very much for your comments. We added additional explanations, in particular, throughout the validation part, but also extended, for example, the summary and discussion related to your in-line comments below. Moreover, we added more details to the radiative transfer simulations for the synthetic*

*data and we extended the existing description of the IDEFAX. For all changes throughout the paper draft, please see the provided latexdiff.*

**In-line comments (many related to the minor revision):**
104, 120, and elsewhere
"Observation of the cloudbow indicates the presence of liquid water and absence of the cloudbow a pure ice cloud." (104)
"If the cloudbow is geometrically possible but not visible, the cloud consists of pure ice and the ice fraction equals to 1." (120)
See major revision above - the Q signal may appear as pure ice, but contain a mix of ACA and ice (in general). This can change the interpretation of ice fraction.
*Thank you very much for noting that. We changed the corresponding sentences to:*
*"Observation of the cloudbow generally indicates the presence of liquid water and absence of the cloudbow a pure ice cloud."*
*"If the cloudbow is geometrically possible but not visible, the cloud is assumed to consist of pure ice and the ice fraction is assumed to be equal to 1."*
*In addition, we also adjusted similar statements. A more detailed discussion about the influence of aerosols is then given later in the paper.*

121
What does it mean for the polarization signal to be "saturated"? As in the top of the detector dynamic range? Or does that mean that the cloud has a COT > ~3 and therefore, "infinite" to a photon? Please explain in-text here.
I realized later on this definition is on line 215 - far too late into the paper. Please bring this up to an earlier section.
*We added an explanation what saturated in this case means and further referred to Sect. 3.4:*
*"Here, cases where the polarization signal of Q is saturated or not saturated are distinguished (see Sect. 3.4). Saturated refers here to a cloud with an optical thickness larger than about 3 to 5, such that the polarization signal is independent of the cloud optical thickness. ..."*

135
How does the Kolling et al. algorithm treat cloud sides/edges? A bit more discussion about this would be great.
*The retrieval of 3D cloud geometry by Kölling et al. (2019) is based on feature detection and stereographic reconstruction. The cloud sides/edges are not treated differently from the rest of the clouds. However, the cloud sides often exhibit sharp features compared to often more spatially homogeneous cloud centers. Thus, the feature detection works particularly well for the cloud sides. On the other hand, small mispointing errors introduce larger absolute errors to the retrieved cloud top heights for the cloud sides due to the typically steeper slopes. A detailed discussion about the uncertainties of the retrieval by Kölling et al. (2019) can be found in Volkmer et al. (2024). We added a reference to the uncertainties and Volkmer et al. (2024) to the text.*

*Volkmer, L., Pörtge, V., Jakub, F., and Mayer, B.: Model-based evaluation of cloud geometry and droplet size retrievals from two-dimensional polarized measurements of specMACS, Atmospheric Measurement Techniques, 17, 1703–1719, https://doi.org/10.5194/amt-17-1703-2024, 2024.*

159

How robust is the minimum checking on Q to instrument measurement noise?

*There is of course some influence from instrument noise. However, we detect a cloudbow depending on the difference in the measured signal depending on the standard deviation of the measurements. So, we at least partly account for the influence of instrument measurement noise. Noise will, however, likely increase the difference between the observed minimum and maximum and therefore rather falsely identify a non-existing cloudbow as a cloudbow. In this case, the computation time is increased because the phase partitioning retrieval has to be performed more often, but the retrieved ice fraction is not affected. Therefore, the cloudbow detection and following phase retrieval should be robust against noise. High instrument noise additionally leads to larger RMSEs and consequently can be detected and filtered from the results if necessary. We added a sentence mentioning the influence of instrument noise to the section about the cloudbow detection (see answer to the next comment below).*

176 (and following paragraph)

I am concerned that manual cloudbow labeling does not accurately represent the true uncertainty of the cloudbow detection, and confuses the interpretation of 3D effects and other errors in the applications later in the paper.

For example, the specMACS Q uncertainty between 3.5-6% given in Weber et al. (2024) could bury weak cloudbows in noise and add error in human interpretation. This could be where the 23.4% false detection metric is coming from.

There is evidence from this and other work (van Diedenhoven et al. (2012), and unpublished from Xu et al. on PACE/HARP2) that the ice/water detection is straightforward with multi-angle polarization statistics. As noted, the high 4% false positive metrics is likely human error as well.

Instead, I recommend a more statistical approach using Qual and RMSE metrics from Portge et al. (2023) to verify the cloudbow detection. Simple thresholds on both could differentiate real cloudbows from noise or false positives. Since the cloudbow Q fit is already part of the flow, aren't these metrics part of the calculation?

It is also valuable to have an extra category "unknown" for cloudbow cases that are ambiguous. There is a precedent for "unknown" in other cloud phase indices (esp. Reidi et al. 2010) and may clarify the results that pass RMSE minimization.

Pörtge, V., Kölling, T., Weber, A., Volkmer, L., Emde, C., Zinner, T., Forster, L., and Mayer, B.: High-spatial-resolution retrieval of cloud droplet size distribution from polarized observations of the cloudbow, Atmos. Meas. Tech., 16, 645–667, https://doi.org/10.5194/amt-16-645-2023, 2023.

van Diedenhoven, B., A. M. Fridlind, A. S. Ackerman, and B. Cairns, 2012: Evaluation of Hydrometeor Phase and Ice Properties in Cloud-Resolving Model Simulations of Tropical Deep Convection Using Radiance and Polarization Measurements. J. Atmos. Sci., 69, 3290–3314, https://doi.org/10.1175/JAS-D-11-0314.1.

Weber, A., Kölling, T., Pörtge, V., Baumgartner, A., Rammeloo, C., Zinner, T., and Mayer, B.: Polarization upgrade of specMACS: calibration and characterization of the 2D RGB polarization-resolving cameras, Atmos. Meas. Tech., 17, 1419–1439, https://doi.org/10.5194/amt-17-1419-2024, 2024.

*Thank you very much for expressing this concern. As mentioned in the paper draft, the purpose of the cloudbow detection is to provide a pre-selection of the data to reduce the computation time of the phase retrieval. With the manual labelling, we wanted to give a rough estimate of the uncertainty of the cloudbow detection. We totally agree that there is an influence of measurement noise and error in human interpretation. Nevertheless, the uncertainty of the cloudbow detection is also included in the total retrieval*

*uncertainty, which we quantified through the detailed validation studies. So, the manual labelling and potential uncertainties related to that do not affect the analyses and discussions later in the paper, but should only provide a rough validation of the cloudbow detection part of the retrieval. We have also thought about other ways how to perform the cloudbow detection and to validate it. This includes thresholds on the quality index and RMSE of the cloudbow retrieval, as suggested by you. However, Veronika Pörtge found in her work that a threshold-based approach on these metrics to filter signals without cloudbow from the cloudbow retrieval results did also not work perfectly. Therefore, she additionally developed a simple random forest classification algorithm to exclude signals without cloudbow from her analyses. This algorithm was, however, developed for observations with high sun in the tropics and requires the entire cloudbow range to be observed, which was not the case for the observations during HALO-(AC)³ in the Arctic, and the labelling of the training data is also influenced by human interpretation. Therefore, we finally decided to use manual labelling of a significant number of cloudbow signals performed by different people. Adding an additional "unknown" category to the labels is a very good idea. However, the manual labelling was time consuming and we would be very happy if we would not have to repeat that. Anyways, we added a discussion about your concerns to the section about the validation of the cloudbow detection:*

*"To reduce personal biases, the manual labeling was done by different people. Nevertheless, the manual labelling is affected by measurement noise and human interpretation. The determined accuracy of the cloudbow detection should, therefore, be interpreted as a rough estimate to prove the general applicability of the introduced cloudbow detection method. A detailed uncertainty assessment of the entire phase retrieval is carried out later in Sect. 5."*

194

Also aerosol optical thickness (see major revision)

*Thank you very much for noting that. We have not added the aerosol optical thickness here, since it is not yet part of the forward operator. However, we added an additional explanation in the discussion section, mentioning that the aerosol optical thickness could and should be included as an additional parameter in the future.*

214

Add to the end "since ice clouds are brighter than liquid clouds, in our simulated cases."

*Changed as suggested.*

221 and elsewhere through the paper

All mentions of "reflectivity" should be "reflectance".

*The second reviewer actually asked us, to refer to reflectivity instead of reflectance. The use of reflectance and reflectivity differs between different publications. Since reflectivity was also used in the already published paper about the IDEFAX, we would like to use reflectivity instead of reflectance to be consistent.*

220

Figures 1 and 3 show that the change in Q at different COD is nowhere near the same magnitude as the change in I, but ice fraction changes to Q happen almost independently to COD.

The consequence of a combined, equally weighted RMSE for I and Q in unsaturated cases is that the "winning solution" for ice fraction may overemphasize a good I comparison over Q, where the distinct information content is.

This may explain why biases in measured vs. modeled ice fraction persist in the Figure 10f histograms for in the cloudbow range retrieval - and also why the COD retrieval compares well on 11f.

I recommend considering an error-normalized metric instead, such as:

$$\chi = \left(1 - w_Q\right) \frac{I_{meas} - I_{model}}{RMSE_{I,meas-model}} + w_Q \frac{Q_{meas} - Q_{model}}{RMSE_{I,meas-model}} \quad\quad (1)$$

where w Q is an empirical weight on Q. This form allows Q to directly compensate for measurement-model differences in I. $w\,Q$ may be effective at 0.5, but may need fine tuning to emphasize the independent information content in Q relative to ice fraction.

*Thank you very much for your suggestion towards an improved metric for the optimization! We also noticed that the changes in I and Q depending on the optical thickness and ice fraction are not of similar magnitude and that the error metric has to be chosen carefully in order to not prefer the optimization of one variable over the other. During the retrieval development, we tested several different metrics, including weighted means of the differences for I and Q, similar to your suggestion above. The product form, we present in the paper was finally chosen since it showed the best results based on test with synthetic data. By using the product form, we avoid the problem of different magnitudes. In addition, we added small numbers to account for cases where one of the errors is very small such that the other value could be chosen arbitrarily. Nevertheless, there might still be more optimal solutions than the one we found. We added some more information about our choice to the corresponding section:*

*"Different error metrics for the optimization were tested, including commonly used weighted sums of the RMSE of I and Q. The product form for the combined RMSE was finally chosen since it showed the best results. Nevertheless, other improved optimization metrics could be tested and incorporated in the future."*

250

Of the two cases shown in Figure 4, neither is labeled as "homogeneously mixed". Do you mean "linearly distributed"?

*There are, in general, two extreme cases for the phase partitioning in Arctic mixed-phase clouds. The first one is a completely homogeneously mixed cloud. In that case, the vertical attribution of the retrieved ice fraction is irrelevant since the ice fraction throughout the cloud is constant and the penetration depth does not matter. The second case is a cloud with completely spatially separated phases with a liquid water layer on top of an ice layer. In this case, the vertical optical thickness to which the signal is sensitive to matters. The two cloud cases shown in Fig. 4 are used to quantify this vertical optical thickness threshold, which would not be possible with a homogeneously mixed cloud.*

Figure 8

The terms "cloudbow" and "slope" for the third column histograms were not immediately obvious. Please describe this more explicitly like:

"(c, f) Histogram of the differences between retrieved and model ice fractions with mean and standard deviation calculated from analysis performed in the cloudbow scattering angle range (blue) and forward scattered slope range (orange)"

And also please harmonize other figures that may have similar discussion.

*Changed as suggested.*

350

I strongly suggest adding 2-panel figure that shows spatially, over the specMACS domain:

- The cloud pixels that correspond to the slope range retrieval, and which ones to the cloudbow range retrieval
- The cloud pixels that undergo the saturated retrieval (Q only) and which ones go through the unsaturated retrieval (I and Q).

I am curious if these distributions can help explain some of the spatial variation in the 3D study row of Figure 8 (d,e,f). This will also support discussion on errors (line 354 - 385).

*Thank you very much for noting that! The information is actually already included in the plots, but it was not discussed in detail so far. The retrieval results shown in the upper left part of panels (a,b,d,e) of Fig. 8 and similar figures correspond to the slope angular range, the results in the lower right part are for the cloudbow range. This information is already given in the figure captions and also mentioned in the text, fo example, during the explanation of Fig. 8. A comparison of Fig. 8 and 9 further shows, which pixels go through the saturated and which through the unsaturated retrieval. Pixels where only the ice fraction was derived correspond to a saturated polarization signal. Pixels with both retrieved ice fraction and optical thickness have unsaturated polarization signals. We added more discussion about that to the text.*

*"... The retrieved values in the lower right of the images correspond to the cloudbow range and the values in the upper left part to the slope range... The optical thickness is only retrieved at the cloud edges, where the clouds are optically thinner and the polarization signal is not saturated. Hence, pixels showing both a retrieved ice fraction and optical thickness in Fig. 8 and 9 have an unsaturated polarization signal and undergone the combined retrieval using I and Q. Pixel with a retrieved ice fraction and without a retrieved optical thickness correspond to pixels with a saturated polarization signal for which the ice fraction is directly derived from Q."*

Summary section

Given that the realistic 3D cloud simulations compare the best against specMACS data - of the four retrieval combinations: unsaturated slope, unsaturated cloudbow, saturated cloudbow, saturated slope - which are the most valuable and which are least effective? It is clear from Figure 13 that they may create different results and it would be excellent to summarize under what conditions they succeed and aren't as useful.

*We added an additional summary concerning the cloudbow and slope angular ranges and the saturated and unsaturated polarization signals to the summary:*

*"Generally, the retrieval using the cloudbow angular range is more accurate than the retrieval using the slope angular range, as the cloudbow range is more sensitive to the cloud thermodynamic phase and more strongly dominated by single scattering, and should therefore be preferred if the observation of the cloudbow is geometrically possible. In addition, the uncertainty of the retrieval results for saturated polarization signals is higher than for unsaturated signals. In the former case, the ice fraction can directly be derived from measurements of Q, whereas in the latter a combined retrieval using I and Q has to be applied. I is more strongly affected by 3D radiative effects than Q, and additionally the signals of I and Q originate from different penetration depths within the cloud, which increases the uncertainty of the derived ice fraction."*